# The synergistic impact of Spirulina and selenium nanoparticles mitigates the adverse effects of heat stress on the physiology of rabbits bucks

Ibrahim T. El-Ratel[1], Mawada E. Elbasuny[1], Hamdy A. El-Nagar[2], Abdel-Khalek E. Abdel-Khalek[3], Ali A. El-Raghi[1], Mohammed Fouad El Basuini [4,5]*, Khaled H. El-Kholy[1], Sara F. Fouda[6]

1 Department of Animal, Poultry and Fish Production, Faculty of Agriculture, Damietta University, Damietta, Egypt, 2 Department of Biotechnology Research, Animal Production Research Institute, Agricultural Research Center, Giza, Egypt, 3 Department of Animal Production, Faculty of Agriculture, Mansoura University, Mansoura, Egypt, 4 Department of Animal Production, Faculty of Agriculture, Tanta University, Tanta, Egypt, 5 Faculty of Desert Agriculture, King Salman International University, South Sinai, Egypt, 6 Department of Poultry Production, Faculty of Agriculture, Mansoura University, Mansoura, Egypt

* m_fouad_islam@yahoo.com

**Data Availability Statement:** All relevant data are within the paper and its Supporting Information files.

## Abstract

Heat stress has a detrimental effect on animal fertility, particularly testicular functions, including reduced sperm output and quality, which causes an economic loss in the production of rabbits. The present trial investigated the efficacy of dietary Spirulina (SP) (*Arthrospira platensis*), selenium nanoparticles (SeNPs), and their combination (SP-SeNPs) on semen quality, haemato-biochemical, oxidative stress, immunity, and sperm quality of heat-stressed (HS) rabbit bucks. Sixty mature bucks (APRI line) were distributed into 6 groups of ten replicates under controlled conditions. Bucks in the 1st group (control-NC) were kept under normal conditions (11–22°C; 40–45% RH% = relative humidity), while the 2nd group (control-HS) was kept under heat stress conditions (32±0.50°C; 60–66% RH %). The control groups were fed a commercial pelleted diet and the other four heat-stressed groups were fed a commercial pelleted diet with 1 g SP, 25 mg SeNPs, 1 g SP+25 mg SeNPs, and 1 g SP+50 mg SeNPs per kg diet, respectively. The dietary inclusion of SP, SeNPs, and their combinations significantly increased hemoglobin, platelets, total serum protein, high-density lipoproteins, glutathione, glutathione peroxidase, superoxide dismutase, and seminal plasma testosterone while decreased triglycerides, total cholesterol, urea, creatinine, and malondialdehyde compared with the control-HS. Red blood cells, packed cell volume, serum albumin, and testosterone significantly increased, while SeNPs, SP+SeNPs25, and SP+SeNPs50 significantly decreased low-density lipoproteins, aspartate, and alanine amino transferees. Total antioxidant capacity substantially increased in serum and seminal plasma, while seminal plasma malondialdehyde decreased in 25 or 50 mg of SeNPs+SP/kg groups. All supplements significantly improved *libido*, sperm livability, concentration, intact acrosome, membrane integrity, total output in fresh semen, and sperm quality in cryopreserved semen. SP-SeNPs50 had higher synergistic effect than SP-SeNPs25 on most

**Funding:** The author(s) received no specific funding for this work.

**Competing interests:** The authors have declared that no competing interests exist.

different variables studied. In conclusion, the dietary inclusion of SP plus SeNPs50 has a synergistic effect and is considered a suitable dietary supplement for improving reproductive efficiency, health, oxidative stress, and immunity of bucks in the breeding strategy under hot climates.

## 1. Introduction

Heat stress has a detrimental effect on animal fertility and causes significant economic losses in rabbit production [1–4]. Oxidative stress performs a crucial function in semen quality, quantity, and fertility of sperm cells. Under heat stress (HS), lipid peroxidation increases, particularly when the temperature and humidity exceed the normal ranges [1, 5]. HS negatively alters testicular functions and causes DNA fragmentation [2], leading to a reduction in the output and quality of spermatozoa [3] and animal well-being [4]. HS increases sperm abnormality by impairing the mitotic division of spermatocytes during spermatogenesis [5]. These negative impacts of HS on bucks are partially associated with generating free radicals [6]. Spermatozoa of rabbits are rich in metabolic activity and poly-unsaturated fatty acids in their plasma membrane; therefore, a rise in free radicals and lipid peroxidation reduces sperm fertilizing ability [1, 7]. Additionally, the consequences of heat stress include oxidative stress, organ damage or dysfunction, endocrine disorders, immunity deficiency, impaired metabolism and energy imbalance, ultimately leading to reduced animal productivity and increased mortality [8, 9].

In order to lessen the harmful impacts of HS and to avoid buck infertility by limiting reactive oxygen species (ROS) production in rabbits, a variety of tools are available. In this way, using natural antioxidants lessens the deleterious impacts of HS on males by assisting with various physiological and metabolic processes [8–13]. Spirulina, a planktonic blue-green alga, exhibits anti-viral, anti-microbial, anti-inflammatory, hepato-protective, and antiparasitic activities and contains antioxidative pigments [14] because it contains a vibrant nutritional profile, particularly proteins, essential amino acids, fatty acids, vitamins, minerals, phytopigments, phycocyanin, polysaccharides, carotenoids, chlorophyll, and phenolic compounds [15–18].

Nano-technology is applied in animal nutrition, and the most basic approaches in this area are nanoparticles (NPs) with 1–100 nm dimensions. The NPs are characterized by higher bioavailability, specific surface area, surface center activity, catalytic and adsorption abilities, and less toxic properties [19]. Therefore, elements in the NPs form at low levels are more efficient than those in conventional forms when added to diets [20]. So NPs can be easily assimilated in the digestive tract [21]. Selenium (Se) is included in several vital physiological processes, like reproductive efficiency, immune status, antioxidant defenses, stress prevention, endocrine, and metabolic processes, and is an essential trace element for animals [22–24]. The Se is a crucial component in the antioxidant enzyme glutathione peroxidase (GSH-Px) [25, 26], which is a significant phase II detoxification enzyme to decrease the level of lipid peroxidation [9]. Dietary Se nanoparticles (SeNPs) supplementation increased the ability of absorption bioavailability, catalytic ability, and surface activity while reducing energy loss [27, 28]. In rabbits, SeNPs provide functional benefits like preventive impacts against reproductive toxicity, enhancing the testosterone level, improving the quality of spermatozoa, and decreasing DNA damage of sperm [29].

SeNPs and/or Spirulina (SP) can be used as promoters of growth, antioxidants, stimulation of immunity, and anti-microbial agents in broilers under HS, whereas their incorporation into the diet can mitigate the negative impacts on the heat-stressed by improving the antioxidant status [16]. Also, dietary supplementation of the SeNPs-SP combination can alleviate the

harmful effects of HS by enhancing the growth efficiency, reproduction, antioxidant status, and immunity of growing and doe rabbits [30, 31]. Furthermore, SP alone was reported to be an effective tool under HS conditions for improving reproduction, stimulating antibodies, producing cytokines, increasing antioxidant capacity, and reducing lipid peroxidation of bucks [9, 32, 33]. Based on these findings, we hypothesize that a combination of SP with different levels of SeNPs may positively affect the reproductive performance of male rabbits under HS.

Thus, this study aimed to explore the effectiveness of dietary Spirulina (SP) (*Arthrospira platensis*), selenium nanoparticles (SeNPs), and their combination (SP-SeNPs) on semen quality, haemato-biochemical, oxidative stress, immunity, and sperm quality of heat-stressed (HS) rabbit bucks.

## 2. Materials and methods

### 2.1 Animals and management protocol

The ethical committee of the Faculty of agriculture at Tanta University approved the experimental protocol and all methods in the present study for treating animals for scientific purposes (Approval No. AY2019-2020/ Session 6/ 2020.01.13). All experiments were performed in accordance with relevant guidelines and regulations. Our reporting of research involving animals follows the recommendations of the ARRIVE guidelines. The study was conducted at a private commercial rabbit farm in Mansoura City, Dakahlia Governorate, Egypt, from December 2021 to April 2022. The experimental animals included 60 sexually mature APRI line bucks (7–8 months of age and 3.15±0.32 kg live body weight). Each buck served as a replicate and was kept individually in stainless steel cage batteries (40×50×35cm) accommodated with feeders and automatic drinkers in a closed system with controlled heat, humidity, and light regimes (12 hrs light: 12 hrs dark) under the same managerial and hygienic conditions. The feeding system of all bucks included a commercial pelleted diet (CPD) based on NRC [34]. The ingredients and chemical composition of CPD are shown in Table 1.

### 2.2 Design of the experiment

The experimental animals (n = 60) were randomly divided into six groups (10/group). Bucks in the 1st group (control-NC, G1) were kept under normal indoor conditions (11–22°C; 40–

**Table 1. Ingredients and chemical composition of the diet (CPD) fed to bucks.**

| Ingredient | % | Chemical analysis | |
|---|---|---|---|
| Berseem hay | 30.05 | Total protein % | 17.75 |
| Barley | 24.60 | Crude fiber % | 12.38 |
| Wheat bran | 21.50 | Total lipids % | 2.27 |
| Soybean meal (44% CP) | 17.50 | Digestible Energy Kcal/Kg diet | 2500 |
| Molasses | 3.00 | Ca % | 1.24 |
| Dicalcium Phosphate (DCP) | 1.60 | Na % | 0.16 |
| Limestone | 0.95 | Total phosphorus % | 0.80 |
| NaCl | 0.30 | Lysine % | 0.98 |
| Premix * | 0.30 | Methionine % | 0.46 |
| DL-Methionine | 0.20 | Methionine + Cystine % | 0.76 |

* 1 kg premix contains: Biotin, 0.05 mg; Choline, 250 mg; Folic acid, 3 mg; Niacin, 50 mg; Pantothenic acid, 10 mg; Vitamin $B_1$, 2 mg; Vitamin $B_{12}$, 0.01 mg; Vitamin $B_6$, 2 mg; Vitamin E, 40 mg; Vitamin $K_3$, 2 mg; Vitamin A, 6000 IU; Vitamin $B_2$, 4 mg; Vitamin $D_3$, 900 IU; Co, 0.1 mg; Cu, 5 mg; Fe, 50 mg; I, 0.2 mg; Mn, 85 mg; Se, 0.1 mg; Zn, 50 mg.

45% RH%) and fed on CPD without supplementation. Bucks in the 2nd group (control-HS, G2) were exposed to HS conditions (32 ± 0.50°C; 60–66% RH) and fed on CPD without supplementation [1]. The other four treatment groups, including bucks kept under HS conditions as in G2 and fed on CPD supplemented with 1g SP (G3), 25 mg SeNPs (G4), 1 g SP+25 mg SeNPs (G5), and 1 g SP+50 mg SeNPs (G6) per kg diet, respectively. The treatment period was two months, followed by a semen collection period of 10 weeks.

Treatment of SP was in powder form (Alga Biotechnology Unit, National Research Center, Dokki, Egypt), and SeNPs treatment (Sigma-Aldrich, USA) were involved in this study. The SeNPs morphology by TEM revealed that the mean particle size of SeNPs was 70 nm and the value of ZP was −4.46 mV which showed the best sign of the potential stability of the colloidal system. The mean polydispersity Index (PDI) was 0.13, indicating that nanoparticles produced by this system had the highest homogeneity in size distribution (Fig 1).

## 2.3 Libido and semen collection

One week before the main semen collection phase (training period), bucks were trained for the semen collection. Sexual desire (libido) was recorded, as a reaction time, as a time in seconds that elapsed from female insertion into the male cage up to the complete ejaculation. Semen was collected by the same veterinarian in the morning (8 a.m.) one/wk. by an artificial vagina (40–41°C), and a teaser doe was used. During the experimental period, semen was collected for 10 successive weeks (7 wks. for fresh semen and another 3 wks. for semen cryopreservation).

## 2.4 Fresh semen evaluation

Semen of 70 ejaculates (10 bucks x 7 weeks) were taken and evaluated in fresh case/group, then an average of each semen parameter/rabbit was calculated. On the day of the semen collection, gel mass (if present) was removed, and net semen volume (NSV) was recorded in the graduated tube connected to the artificial vagina. The pH value was measured with a pH pen. In each semen sample, the score of mass motility (0–5) was determined; the percentage of forward sperm motility (progressive motility) was examined for about 200 sperm cells in five microscopic fields by using a phase-contrast microscope with a hot stage at 37°C (Leica DM 500, Leica Mikrosysteme Vertrieb GmbH, Wetzlar, Germany).

Percentages of sperm morphological abnormality (head and tail) and livability (live: unstained cells and dead: purple stained cells) were determined in about 200 sperm cells stained by eosin-nigrosine dual stain (5%:10%) using a light microscope at 100 × magnification. Semen was diluted with saline solution (1:99) to determine sperm cell concentration (SCC) with a Neubauer hemocytometer slide (GmbH +Co., Brandstwiete 4, 2000 Hamburg 11, Germany).

The hypo-osmotic swelling test (HOS-t) was employed to measure sperm membrane integrity by incubation (37°C for one hour) of 30 μl semen sample and 300 μl a solution with osmolarity level of 100 mOsm/kg) containing fructose (9 g) and sodium citrate (4.9 g) in 100 ml of distilled water in a water bath [6]. Responses of about 200 spermatozoa to be swollen or with curled tails were determined, and membrane integrity was calculated. Total sperm outputs (TSO) per ejaculate was computed according to the following equations: TSO/ejaculate ($10^6$/ejaculate) = NSV (ml) x SCC (×106/ml). Acrosomal status, including sperm cells with intact acrosome cap, non-intact acrosome, and sperm cells without acrosome cap, were determined to calculate the percentage of intact acrosome using naphthol yellow S and erythrosine B [1].

## 2.5 Semen cryopreservation

In cryopreserved semen (30 ejaculates/group) collected from 10 bucks/group during the last 3 weeks of the experiment, ejaculates without gel mass of each group were pooled, diluted (1

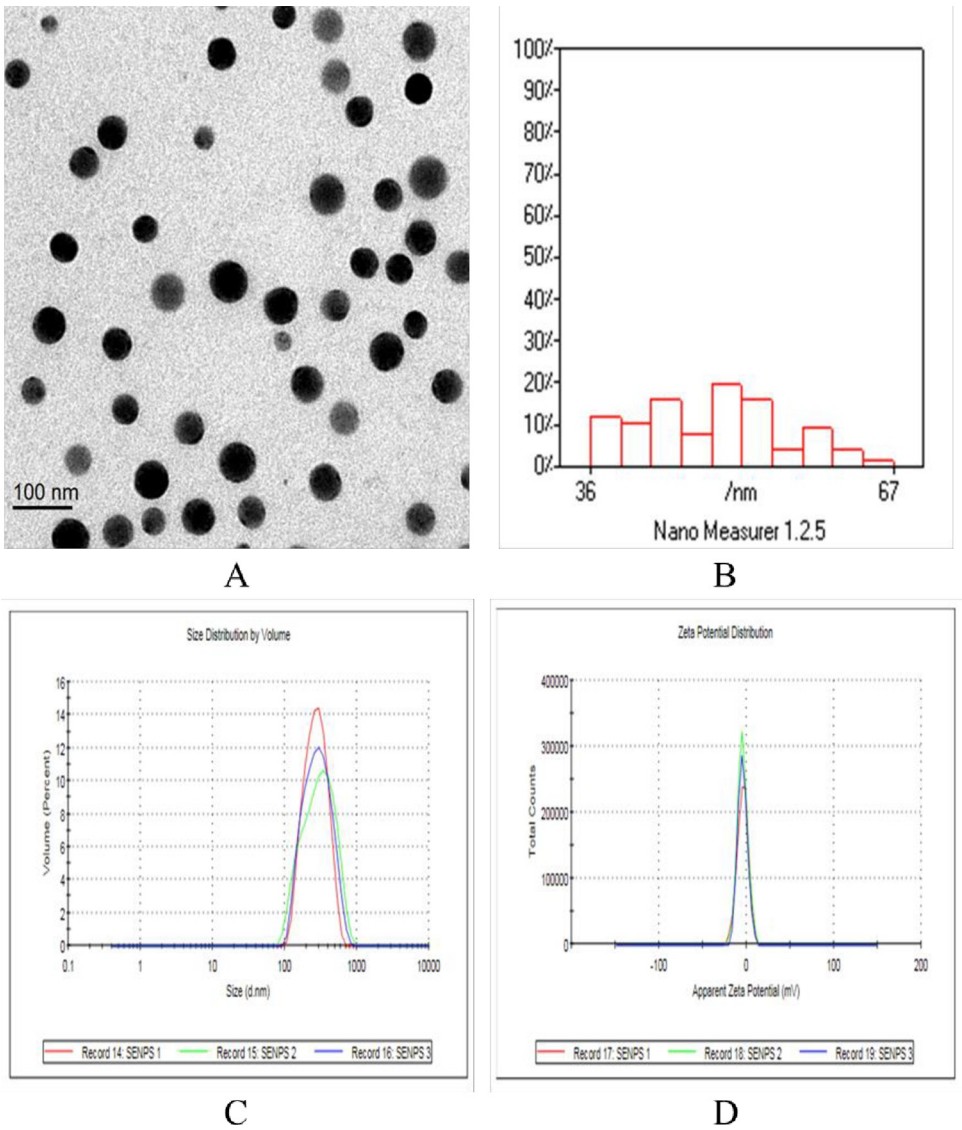

**Fig 1.** (A) Transmission electron microscopy morphology of SeNPs; (B) Histogram of particles size; (C) Transmission electron microscopy size distribution; (D) Zeta potential.

semen: 5 diluents) with Tris-extender. Each 100 ml of the extender contained Tris (3.028 mg), citric acid (1.675 mg), fructose (1.250 mg), gentamycin (5000 μg), 20% egg yolk, 7% glycerin, and distilled water up to 100 ml. Semen was packed in 0.25 mL straws (IVM technologies, L' Aigle, France) after the dilution process and equilibration (at 5˚C for 4 h), then offered at 4–5 cm overhead liquid nitrogen (LN) for 10 min, then preserved in LN at -196˚C for 30 days.

Cryopreserved semen straws were thawed at 37˚C for 30 s in a water bath. Post-thawed semen was evaluated for sperm forward motility, livability, abnormality, and membrane integrity as previously conducted in semen at the fresh case.

## 2.6 Analytical procedures

**2.6.1 Blood variables.** At the end of the experimental period, 5 bucks in each treatment were randomly selected for blood testing. Blood samples were gathered from the marginal ear-

vein of the animals after topical anesthetized by Xylocaine 4% anesthetic into heparinized pipes to assess hematological parameters. Other blood samples were collected into non-heparinized tubes and left to clot, then centrifuged at 700 $g$ for 20 min at 4˚C to separate serum collection, which was stored at −20˚C until subsequent analyses of blood biochemicals and antioxidant and immunity markers. Hematological parameters included hemoglobin concentration (Hb), packed-cell volume (PVC), and counts of red blood cells (RBCs), white blood cells (WBCs), and platelets using HB 7021 analyzer. Blood serum biochemical included concentrations of total protein (TP), albumin (AL), triglycerides (TG), total cholesterol (TC), high (HDL) and low-density (LDL) lipoproteins, urea, and creatinine as well as activity of aspartate (AST) and alanine (ALT) amino transferees. Globulin was determined by calculation (the differences between concentration of TP and AL. In blood serum, levels of total antioxidant capacity (TAC), glutathione (GSH), glutathione peroxidase (GPx), superoxide dismutase (SOD), and malondialdehyde (MDA) were assayed. All biochemicals, enzyme activity, and oxidative capacity studied were assayed colorimetrically by Bio-diagnostic Co. kits (wwwBio-diagnostic.com) and UV-VIS Auto spectrophotometer (UV-2602, Labomed, Los Angeles, CA, USA). Also, enzyme-immunoassay of serum testosterone profile was employed using commercial kits (Biosource-Europe S.A. 8, rue de L'Lndustrie. B-1400 Nivelles, Belgium). Coefficients of intra- and inter-assay were 7.8 and 8.4%, respectively. The minimum and maximum detectable limits were 0.1 and 18.0 ng/ml, respectively.

**2.6.2 Seminal plasma variables.** Samples of fresh semen collected at the last week of the semen collection (the 7[th] week) from 10 bucks per group were centrifuged (700 $g$ for 20 min) to obtain the seminal plasma, which was stored at –20˚C) for determining TAC, MDA, and testosterone. Protein carbonyl levels (POC) in sperm media of the post-thawed semen collected during the last semen collection period (the 10[th] week) were also determined. POC was determined with MBS2600784 protein carbonyl ELISA kit (MyBioSource, Giza, Egypt) based on the manufacturer's technique [35].

**2.6.3 Evaluation of sperm fertility in bucks.** Sperm fertility of fresh semen of each group was performed by using 120 sexually mature APRI rabbit females. All does were divided into 6 groups (n = 20), then does in each group naturally mated with 5 bucks/treatment (4 does/buck) during 3 days as a mating interval at the end of the experiment. On days 10–12 post-mating, manual abdominal palpation was performed to determine the pregnancy rate (PR). PR = (positive does (n)/mated does (n) ×100. After parturition, the kindling rate (KR) as numbers of kindled does divided by the number of positive does ×100 was calculated. Also, the litter size of does and viability rate of kits (total born and alive) were calculated. At weaning (28 days of kit age), litter size and viability rates (live and dead) were recorded.

## 2.7 Statistical analysis

Statistical analysis comprised the General Linear Model procedures using SAS (Cary, NC, USA: SAS Institute Inc, 2012) to study the effect of treatment by the GLM model: $Y_{ij} = \mu + T_i + e_{ij}$, where $Y_{ij}$, observations; $\mu$, overall mean; $T_i$, effect of treatments (i, 1–6); $e_{ij}$, random error. Tukey's Studentized Range test was used to separate the differences between treatment groups at $P < 0.05$.

## 3. Results

### 3.1 Blood hematology and serum biochemistry

Data in Table 2 illustrate the impacts of SP, SeNPs, and their combinations on blood hematology and serum biochemistry. Significant alterations (p < 0.01) in all blood hematology parameters were observed except WBCs (p = 0.1538). In comparing control-NC and control-HS

**Table 2. Effect of dietary supplemental Spirulina, selenium nanoparticles, and their combinations on hematological and biochemical parameters of rabbit bucks under heat stress conditions.**

| Items | Control-NC | Treatments (mg/kg diet) | | | | | *P*-value |
|---|---|---|---|---|---|---|---|
| | | Control-HS | SP | SeNPs | SP+SeNPs25 | SP+SeNPs50 | |
| **Hematological variables** | | | | | | | |
| Hemoglobin (g/dl) | 12.5±0.08[a] | 9.18±0.05[d] | 11.93±0.04[b] | 10.5 ±0.14[c] | 12.19 ±0.09[ab] | 12.28± 0.03[ab] | <0.001 |
| Red blood cells ($10^6$/mm$^3$) | 6.81±0.09[a] | 5.88± 0.2[c] | 6.10 ±0.06[bc] | 6.26±0.03[b] | 6.35± 0.03[b] | 6.36 ±0.02[b] | <0.001 |
| White blood cells ($10^3$/mm$^3$) | 7.38±0.05 | 7.08 ± 0.05 | 7.19± 0.02 | 7.26± 0.03 | 7.25± 0.02 | 7.19± 0.02 | 0.1538 |
| Platelets ($10^3$/mm$^3$) | 253.1±1.5[a] | 205.02±3.1[c] | 219.24± 3.1[b] | 221.12±3.7[b] | 226.7±1.01[b] | 224.23± 2.2[b] | < .0001 |
| Packed-cell volume (%) | 39.6±1.1[a] | 27.8± 1.5[d] | 30.3±0.93[cd] | 32.9±1.02[bc] | 34.6±0.7[b] | 33.48± 1.4[bc] | < .0001 |
| **Blood metabolites** | | | | | | | |
| TP (g/dl) | 7.02±0.06[a] | 6.13 ±0.08[c] | 6.28 ±0.03[b] | 6.31± 0.04[b] | 6.40 ± 0.02[b] | 6.41± 0.02[b] | <0.001 |
| Albumin (g/dl) | 4.01±0.034[a] | 3.21±0.015[c] | 3.25±0.02[bc] | 3.26± 0.024[b] | 3.31±0.023[b] | 3.30±0.01[b] | <0.001 |
| Globulin (g/dl) | 3.02±0.09[ab] | 2.92± 0.08[b] | 3.03 ±0.02[ab] | 3.04 ± 0.03[ab] | 3.09 ±0.03[ab] | 3.11± 0.03[a] | <0.001 |
| Triglycerides (mg/dl) | 56.61±1.36[d] | 86.24±1.711[a] | 78.27± 1.15[b] | 79.42 ±0.83[b] | 66.68 ±0.54[c] | 67.95±0.78[c] | <0.001 |
| Total cholesterol (mg/dl) | 83.08±1.6[c] | 114.29 ± 1.8[a] | 97.50±1.7[b] | 96.83± 3.3[b] | 81.98±2.1[c] | 82.95±2.3[c] | <0.001 |
| High-density lipoproteins (mg/dl) | 35.67±1.4[a] | 22.17 ± 1.1[d] | 26.23± 1.3[c] | 28.88± 1.3[cb] | 31.86± 1.3[ab] | 32.13±1.6[ab] | <0.001 |
| Low-density lipoproteins (mg/dl) | 38.95 ±1.1[b] | 50.47 ± 2.3[a] | 47.94 ±2.4[a] | 42.35±1.3[b] | 41.42±1.3[b] | 40.16±1.4[b] | 0.0004 |
| Aspartate amino transferase (IU) | 23.65±1.1[c] | 34.32±1.3[a] | 30.23±1.6[ab] | 29.27 ± 1.3[b] | 26.66±1.8[bc] | 27.69±1.4[bc] | 0.0007 |
| Alanine aminotransferase (IU) | 20.05±1.8[c] | 28.09±1.3[a] | 26.48 ±0.9[ab] | 23.46 ± 1.4[bc] | 22.73 ±0.95[cb] | 23.15±0.75[bc] | 0.0026 |
| Urea (mg/dl) | 28.04±1.5[c] | 39.58 ± 1.9[a] | 34.48±1.2[b] | 34.04 ±1.7[b] | 31.75 ±1.9[bc] | 31.05 ±1.6[bc] | 0.0012 |
| Creatinine (mg/dl) | 1.06±0.023[d] | 1.41±0.047[a] | 1.26 ±0.025[b] | 1.18±0.027[bc] | 1.19±0.041[bc] | 1.15±0.042[cd] | < .0001 |

Control-NC: control under normal conditions, Control-HS: control under heat stress conditions; SP: spirulina; SeNPs: selenium nanoparticles; SP+SeNPs25: 1g SP + 25 mg SeNPs; SP+SeNPs50: 1g SP+50 mg SeNPs

Values with distinctive superscripts within a row are varied ($P < 0.05$)

groups, the results indicated the harmful effects of HS on all hematological and biochemical variables studied. In comparison with the control-HS group, all treatments increased Hb concentration and platelet count (P<0.05). The counts of RBCs and PCV were increased in SeNPs, SP+SeNPs25, and SP+SeNPs50 groups.

All serum biochemical parameters were statistically affected (p<0.001) by the dietary SeNPs enriched with SP (Table 2). All treatments substantially (p<0.05) boosted serum total protein and HDL, while significantly (P<0.05) decreased TG, TC, urea, and creatinine concentration as compared to the control-HS. The concentration of serum AL increased (P<0.0), while LDL, AST, and ALT decreased (P<0.05) in SeNPs, SP+SeNPs25, and SP+SeNPs50 as compared to the control-HS and SP groups. The concentration of serum GL significantly (P<0.05) increased in the SP+SeNPs50 group compared to other groups. The control-HS group exhibited the poorest values of all parameters studied above compared to the treated and control-NC groups.

## 3.2 Oxidative status in serum and seminal plasma

The impacts of the dietary Spirulina, selenium nanoparticles, and their interaction on redox status in serum are revealed in Table 3. Antioxidant capacity markers were considerably (P<0.05) lower in the control-HS group than in the control-NC group, indicating the effects of HS on increasing lipid peroxidation in serum and seminal plasma. All dietary supplementations to buck diets increased (P<0.05) GSH, GPx, and SOD in serum, while decreasing (P<0.05) MDA level as compared to the control-HS. Enriching rabbit diets with 25 or 50 mg of SeNPs plus SP/kg diet produced significantly (P<0.05) the highest levels of TAC compared

**Table 3. Effect of dietary supplemental Spirulina, selenium nanoparticles and their combinations on oxidative biomarkers in blood and seminal plasma of rabbit bucks under heat stress conditions.**

| Items | Control-NC | Treatments (mg/kg diet) | | | | | P-value |
|---|---|---|---|---|---|---|---|
| | | Control-HS | SP | SeNPs | SP+SeNPs25 | SP+SeNPs50 | |
| **Oxidative markers in the blood** | | | | | | | |
| Total antioxidant capacity (nmol/L) | 1.94 ± 0.03[a] | 0.86 ±0.04[c] | 0.95±0.02[bc] | 0.98± 0.05[bc] | 1.22± 0.08[b] | 1.36± 0.08[b] | < .0001 |
| glutathione (mg/dl) | 19.5±0.8[a] | 9.3± 0.5[d] | 12 ± 0.4[c] | 13.34 ±0.4[c] | 15.3±0.8[b] | 15.3±0.5[b] | < .0001 |
| glutathione peroxidase (IU) | 3.53 ± 0.14[a] | 1.42±0.08[d] | 2.06± 0.08[c] | 2.28± 0.09[bc] | 2.37± 0.03[b] | 2.44± 0.04[b] | < .0001 |
| superoxide dismutase (IU) | 32.3± 1.04[a] | 19.4± 0.8[c] | 22.8± 0.6[b] | 23.6± 1.6[b] | 23.8±0.5[b] | 24.5± 0.8[b] | < .0001 |
| malondialdehyde (nmol/mL) | 4.29± 0.3[c] | 8.13±0.5[a] | 6.24± 0.1[b] | 6.22± 0.1[b] | 6.17± 0.1[b] | 6.14± 0.05[b] | < .0001 |
| **Oxidative markers in the seminal plasma** | | | | | | | |
| Total antioxidant capacity (nmol/L) | 22.65± 1.3[a] | 10.01± 1.8[c] | 10.62±1.02[c] | 12.85±0.68[bc] | 15.1±0.82[b] | 15.24±0.71[b] | < .0001 |
| malondialdehyde (nmol/mL) | 35.06±2.9[b] | 51.88±1.9[a] | 50.56±1.7[a] | 46.4± 1.2[ab] | 45.91± 1.5[b] | 45.8± 1.05[b] | 0.0002 |

Control-NC: control under normal conditions, Control-HS: control under heat stress conditions; SP: spirulina; SeNPs: selenium nanoparticles; SP+SeNPs25: 1g SP + 25 mg SeNPs; SP+SeNPs50: 1g SP+50 mg SeNPs. Values with distinctive superscripts within a row are varied ($P < 0.05$)

to other groups. Concerning the antioxidant activities in the seminal plasma of bucks, TAC increased ($p < 0.05$). At the same time, the MDA level decreased (P<0.05) by feeding rabbit diets with 25 or 50 mg of SeNPs plus SP/kg compared to other groups. The control-HS group exhibited the lowest activities of all the above antioxidant markers compared to the treated and control-NC.

### 3.3 Libido and semen production

The libido of rabbit bucks was improved (p<0.05) by the dietary treatments compared to the control-HS group but did not differ from that of the control-NC. Mass motility score and progressive motility percentage significantly (P<0.05) increased by SeNPs or their combination with SP compared to the control-HS and SP groups. All supplements significantly (P<0.05) increased sperm livability, SCC, intact acrosome, membrane integrity, and TSO while decreasing abnormality compared with the control-HS group. The differences in NSV among the experimental groups were not significant. The most improvement in semen characteristics was by SP+SeNPs50, but still considerably (P<0.05) lower than the control-NC, except RT and mass motility (Table 4).

### 3.4 Sperm quality in post-thawed semen

The effects of dietary treatment supplements on sperm properties in post-thawed semen are presented in Table 5. The present results indicated that the dietary SP, SeNPs, and their combination supplements significantly ($P$ <0.05) enhanced sperm quality in cryopreserved semen in terms of increasing SPM, SL, and MI and decreased sperm abnormality and PCO in comparison with the control-HS. It is of interest to note the maximum improvement occurred in sperm characteristics after thawing by SP+SeNPs50 as compared to the control-HS still to be lower than that exhibited in the control-NC group.

### 3.5 Testosterone in serum and seminal plasma

Results illustrated in Table 6 showed that testosterone level was significantly ($P$ <0.05) increased by SeNPs and their combinations in blood serum with SP, while significantly ($P$ <0.05) increased in the seminal plasma by all supplements.

**Table 4. Effect of dietary supplemental Spirulina, selenium nanoparticles and their combinations on semen quality parameters of rabbit bucks under heat stress conditions.**

| Items | Control-NC | Treatments (mg/kg diet) | | | | | |
| --- | --- | --- | --- | --- | --- | --- | --- |
| | | Control-HS | SP | SeNPs | SP+SeNPs25 | SP+SeNPs50 | P-value |
| Reaction time (sec.) | 14.0 ±0.7[c] | 23.7 ± 0.6[a] | 20.8±1.3[b] | 19.2±1.1[b] | 14.3±0.7[c] | 13.9±0.6[c] | < .0001 |
| Net semen volume (ml) | 0.75 ±0.05 | 0.55±0.02 | 0.66±0.02 | 0.64±0.03 | 0.65±0.03 | 0.67±0.02 | 0.069 |
| Mass motility | 3.8±0.2[a] | 2.6±0.16[c] | 2.9±0.2[bc] | 3.3±0.15[ab] | 3.5±0.17[a] | 3.5±0.17[a] | 0.0002 |
| Progressive motility (%) | 82.5±2.0[a] | 55 ± 2.6[d] | 75±3.4[cd] | 63.5±2.6[bc] | 66.5±3.3[b] | 70.5± 2.3[b] | < .0001 |
| Sperm livability (%) | 85.9±1.8[a] | 51.1±1.6[e] | 63.4±1.03[d] | 68.5± 1.7[c] | 71.7±1.17[bc] | 72.8±0.8[b] | < .0001 |
| Sperm abnormality (%) | 11.8 ±0.9[c] | 24.9±0.9[a] | 21.2±1.05[b] | 20.8±1.05[b] | 20.4± 1.7[b] | 18.2±0.9[b] | < .0001 |
| Sperm cell concentration ($10^6$/ml) | 356.1±2.2[a] | 196.5±5.7[f] | 224.8± 3.2[e] | 254.3±3.5[d] | 285.5±3.9[c] | 296.8±2.6[b] | < .0001 |
| Intact acrosome (%) | 89.4 ±1.6[a] | 60.9±1.04[e] | 74.8± 1.2[d] | 73.7±0.9[d] | 78.2± 0.8[c] | 83.8±0.9[b] | < .0001 |
| Membrane integrity (%) | 45.1 ±1.4[a] | 21.3 ±0.7[d] | 26.1±0.9[c] | 26.7± 0.6[c] | 36.7± 1.04[b] | 37.1±0.8[b] | < .0001 |
| Total Sperm output ($10^6$/ejaculate) | 266.6±17.9[a] | 107.4±4.9[e] | 148.7±4.3[d] | 162.8± 6.1[cd] | 185.1±9.1[bc] | 200.09±6.7[b] | < .0001 |

Control-NC: control under normal conditions, Control-HS: control under heat stress conditions; SP: spirulina; SeNPs: selenium nanoparticles; SP+SeNPs25: 1g SP + 25 mg SeNPs; SP+SeNPs50: 1g SP+50 mg SeNPs. Values with distinctive superscripts within a row are varied ($P < 0.05$)

## 3.6 Sperm fertility

The dietary treatment of bucks used in natural mating of does with both SP plus SeNPs significantly (p<0.05) increased pregnancy rate (PR) and delivered rate (DR) as compared to the control-HS other treatments, but PR and DR were similar to that in the control-NC group. SeNPs and their combinations with SP significantly (P<0.05) improved litter size at birth (Total and live) and weaning, and viability rate *only* at birth (Table 7).

## 3.7 Synergistic impacts of SP-SeNPs versus SP or SeNPs alone

Table 8 shows the percentage of changes in haemato-biochemical, antioxidants, oxidative markers, semen production, testosterone, and reproductive performance of the experimental rabbits. The SP-SeNPs combination had synergistic effects on most variables studied compared to the average of SP or SeNPs alone. The rate of change was higher for SP-SeNPs25 than SP-SeNPs50 on increasing PCV and decreasing levels of triglycerides, cholesterol, LDL, AST, and ALT. However, the rate of change was higher for SP-SeNPs50 than SP-SeNPs25 on increasing Hb, HDL, TAC in serum and seminal plasma, and serum GPx, sperm livability, intact acrosome, and membrane integrity in fresh and thawed semen, sperm concentration and total sperm output in fresh semen, progressive motility in thawed semen, testosterone concentration in serum and seminal plasma, and all reproductive performance of inseminated

**Table 5. Effect of dietary supplemental Spirulina, selenium nanoparticles and their combinations on sperm variables in post-thawed heat-stressed rabbit semen.**

| Items | Control-NC | Treatments (mg/kg diet) | | | | | |
| --- | --- | --- | --- | --- | --- | --- | --- |
| | | Control-HS | SP | SeNPs | SP+SeNPs25 | SP+SeNPs50 | P-value |
| Progressive motility (%) | 72.4± 1.2[a] | 35.1±0.8[e] | 41.2±1.1[d] | 55.5±1.5[c] | 56.4±1.5[c] | 60.9±1.4[b] | < .0001 |
| Sperm livability (%) | 68.9±1.4[a] | 42.3±0.9[d] | 54.6±0.6[c] | 54.6±1.4[c] | 61.5±0.8[b] | 62.8±0.8[b] | < .0001 |
| Sperm abnormality (%) | 13.1±0.7[d] | 26.9± 0.8[a] | 24.5± 0.7[b] | 22.1±0.9[b] | 21.8±1.1[cb] | 21.5±1.2[c] | < .0001 |
| Membrane integrity (%) | 73.5±0.8[a] | 40.7±0.9[e] | 47.4±1.03[d] | 56.8±1.2[c] | 58.2±1.1[c] | 62.30±1.09[b] | < .0001 |
| protein carbonyl (ng/mg) | 2.38± 0.04[c] | 4.04±0.06[a] | 3.80± 0.05[b] | 3.66± 0.12[b] | 3.63± 0.06[b] | 3.58 ± 0.08[b] | < .0001 |

Control-NC: control under normal conditions, Control-HS: control under heat stress conditions; SP: spirulina; SeNPs: selenium nanoparticles; SP+SeNPs25: 1g SP + 25 mg SeNPs; SP+SeNPs50: 1g SP+50 mg SeNPs. Values with distinctive superscripts within a row are varied ($P < 0.05$)

**Table 6. Effect of dietary supplemental Spirulina, selenium nanoparticles and their combinations on blood and seminal testosterone of rabbit bucks under heat stress conditions.**

| Testosterone (ng/ml) | Control-NC | Treatments (mg/kg diet) | | | | | |
|---|---|---|---|---|---|---|---|
| | | Control-HS | SP | SeNPs | SP+SeNPs25 | SP+SeNPs50 | P-value |
| Serum | 3.65±0.053[a] | 2.09 ± 0.029[c] | 2.15±0.052[c] | 2.26±0.019[b] | 2.26±0.016[b] | 2.29±0.017 [b] | < .0001 |
| Seminal Plasma | 3.18±0.033[a] | 2.02±0.067[d] | 2.05±0.057[cd] | 2.17±0.024[bc] | 2.21± 0.019 [b] | 2.26±0.024[b] | < .0001 |

Control-NC: control under normal conditions, Control-HS: control under heat stress conditions; SP: spirulina; SeNPs: selenium nanoparticles; SP+SeNPs25: 1g SP + 25 mg SeNPs; SP+SeNPs50: 1g SP+50 mg SeNPs. Values with distinctive superscripts within a row are varied (*P* < 0.05).

does. On the other hand, the rate of change was higher for SP-SeNPs50 than SP-SeNPs25 on decreasing LDL, urea, creatinine, reaction time, sperm abnormality in fresh and thawed semen, and protein carbonyl in thawed semen.

## 4. Discussion

Heat stress has an undesirable result on the health and fertility of animals and leads to substantial economic losses in the production of rabbits [2–5]. In subtropical regions, the harmful effects of HS on livestock are not easy to mitigate, especially when the high ambient temperature is combined with high humidity. In the present study, the temperature–humidity index (THI) indicated that rabbits in treatment and control-HS groups are exposed to severe heat stress (28.9 to <30.0) [36]. The positive effects of SP or Se in diets of rabbits or broilers under HS conditions, and dietary SP supplementation, as a functional component, due to its high bio-active components and its functional properties have previously been reported [37, 38]. To our knowledge, the present study is the 1st to illustrate the cooperation of SP with different levels of SeNPs on the semen characteristics of heat-stressed bucks.

In the current study, the dramatic reduction observed in the quality and function of buck spermatozoa by HS was reported by El-Ratel et al. [39]. HS condition inhibits hypothalamic gonadotropin-releasing hormone (GnRH) synthesis and secretion. The function of the testis, testosterone production, and the quality of rabbit semen were mainly affected by GnRH [40]. As proved in our study, HS negatively influences the motility and membrane integrity of bucks in the control-HS compared with the control-NC group due to the ROS generation in the testicular tissues of bucks, leading to lipid peroxidation (antioxidant defense system

**Table 7. Reproductive performance of rabbits does mate by rabbit bucks treated Spirulina, selenium nanoparticles, and their combinations under heat stress conditions.**

| Items | Control-NC | Treatments (mg/kg diet) | | | | | |
|---|---|---|---|---|---|---|---|
| | | Control-HS | SP | SeNPs | SP+SeNPs25 | SP+SeNPs50 | P-value |
| Total mated rabbit does (n) | 30 | 30 | 30 | 30 | *30* | 30 | – |
| Pregnant rabbit does (n, %) | 27/30 (90)[a] | 17 (56.67)[b] | 19 (63.33)[b] | 19 (63.33)[b] | 25 (83.33)[a] | 26 (86.67)[a] | – |
| Delivered rabbit does (n, %) | 27/27 (100)[a] | 13 (76.47)[b] | 15 (78.94)[b] | 16 (84.21)[ab] | 23 (92.00)[a] | 25 (96.15)[a] | – |
| Total litter size at birth (n) | 9.04 ±0.37[a] | 5.54±0.38[e] | 5.87± 0.27[de] | 6.81±0.41[cd] | 7.65±0.27[bc] | 8.04±0.23[b] | < .0001 |
| Live litter size at birth (n) | 8.7±0.3[a] | 4.3±0.31[d] | 5.07±0.35[d] | 6.0± 0.4[c] | 6.96± 0.24[b] | 7.44± 0.24[b] | < .0001 |
| Viability rate at birth (%) | 96.7± 1.1[a] | 78.8± 4.6[c] | 85.44±3.5[bc] | 88.5± 2.6[b] | 91.4± 1.8[ab] | 92.68±1.9[ab] | – |
| Litter size at weaning (n) | 8.33 ±0.33[a] | 3.92± 0.33[d] | 4.67± 0.29[d] | 5.63± 0.33[c] | 6.65± 0.25[b] | 7.12± 0.21[b] | < .0001 |
| Viability rate at weaning (%) | 95.80± 1.3 | 91.79±3.05 | 93.75± 2.86 | 94.916± 2.16 | 95.70 ± 1.42 | 96.21± 1.30 | – |

Control-NC: control under normal conditions, Control-HS: control under heat stress conditions; SP: spirulina; SeNPs: selenium nanoparticles; SP+SeNPs25: 1g SP + 25 mg SeNPs; SP+SeNPs50: 1g SP+50 mg SeNPs. Values with distinctive superscripts within a row are varied (*P* < 0.05).

**Table 8. The percentage of change in both levels due to the synergistic impact of the SP and SP-SeNPs.**

| Items | Average | Change % | |
|---|---|---|---|
| | | SP+SeNPs25 | SP+SeNPs50 |
| Hematological variables | | | |
| Hemoglobin (g/dl) | 11.22 | 8.69 | 9.50 |
| Packed-cell volume (%) | 31.60 | 9.49 | 5.95 |
| Blood metabolites | | | |
| Triglycerides (mg/dl) | 78.85 | -15.43 | -13.82 |
| Total cholesterol (mg/dl) | 97.17 | -15.63 | -14.63 |
| High-density lipoproteins (mg/dl) | 27.56 | 15.62 | 16.60 |
| Low-density lipoproteins (mg/dl) | 45.15 | -8.25 | -11.04 |
| Aspartate amino transferase (IU) | 29.75 | -10.39 | -6.92 |
| Alanine aminotransferase (IU) | 24.97 | -8.97 | -7.29 |
| Urea (mg/dl) | 34.26 | -7.33 | -9.37 |
| Creatinine (mg/dl) | 1.22 | -2.46 | -5.74 |
| Antioxidant and oxidative markers in the blood | | | |
| Total antioxidant capacity (nmol/L) | 0.97 | 26.42 | 40.93 |
| glutathione peroxidase (IU) | 2.17 | 9.22 | 12.44 |
| superoxide dismutase (IU) | 23.20 | 2.59 | 5.60 |
| Antioxidant and oxidative markers in the seminal plasma | | | |
| Total antioxidant capacity (nmol/L) | 11.74 | 28.67 | 29.87 |
| Semen production parameters | | | |
| Reaction time (sec.) | 20.00 | -28.50 | -30.50 |
| Sperm livability (%) | 65.95 | 8.72 | 10.39 |
| Sperm abnormality (%) | 21.00 | -2.86 | -13.33 |
| Sperm cell concentration ($10^6$/ml) | 239.55 | 19.18 | 23.90 |
| Intact acrosome (%) | 74.25 | 5.32 | 12.86 |
| Membrane integrity (%) | 26.40 | 39.02 | 40.53 |
| Total Sperm output ($10^6$/ejaculate) | 155.75 | 18.84 | 28.47 |
| Sperm variables in post-thawed | | | |
| Progressive motility (%) | 48.35 | 16.65 | 25.96 |
| Sperm livability (%) | 54.60 | 12.64 | 15.02 |
| Sperm abnormality (%) | 23.30 | -6.44 | -7.73 |
| Membrane integrity (%) | 52.10 | 11.71 | 19.58 |
| Protein carbonyl (ng/mg) | 3.73 | -2.68 | -4.02 |
| Blood and seminal testosterone | | | |
| Blood Serum | 2.21 | 2.49 | 3.85 |
| Seminal Plasma | 2.11 | 4.74 | 7.11 |
| Reproductive performance of rabbits does | | | |
| Pregnant rabbit does (%) | 63.33 | 31.58 | 36.85 |
| Delivered rabbit does (%) | 81.58 | 12.78 | 17.87 |
| Total litter size at birth (n) | 6.34 | 20.66 | 26.81 |
| Live litter size at birth (n) | 5.54 | 25.75 | 34.42 |
| Viability rate at birth (%) | 86.97 | 5.09 | 6.57 |
| Litter size at weaning (n) | 5.15 | 29.13 | 38.25 |

SP+SeNPs25 = 1g SP + 25 mg SeNPs; SP+SeNPs50 = 1g SP+50 mg SeNPs.

damage) in the sperm plasma membrane [41]. The obtained results indicated the deleterious impact of HS on reducing testosterone secretion in bucks of control-HS group via a reduction in testosterone production as a result of destroying the function of Leydig cells [42]. The RT is controlled by the hypothalamus–pituitary–testis axis, nervous system, hormones, and sexual pheromones [43]. Therefore, enhancing testosterone biosynthesis [2, 32] improved the sexual desire of bucks in all treatment groups by decreasing RT compared with the control-HS group. Testosterone is required for spermatogenesis and maturation of sperm cells [44], leading to a marked reduction in abnormality and acrosomal damage. In association with enhancing the testosterone profile, antioxidant administration (SP, SeNPs, or their combinations) improved, quantitatively and qualitatively, the semen production, including mass motility score, and the percentages of sperm progressive motility, livability, normality, acrosome intact and integrity of membrane, SCC, and TSO which adversely affected by HS in rabbits [1, 2]. The pronounced improvement in semen characteristics in our study was by SP+SeNPs50, but still lower (P<0.05) than those of bucks in the control-NC, except RT and mass motility score. In this context, a water supplementation of SeNPs significantly improved semen characteristics in rabbit bucks [27]. Se supplementations had beneficial actions via increasing serum testosterone concentrations, enhanced semen quality, and reduced sperm DNA damage in rabbit bucks [45]. These improvements may be due to the structure and bioavailability of SeNPs molecules versus the inorganic form of Se to cover the needs of the testis and epididymal segments from Se required for spermatogenesis, consequently, high-fertile sperm production and high-quality semen [27]. Supplementation of organic Se also enhanced the quality of semen and seminal plasma of heat-stressed rabbit bucks by improving the accessory sex gland functions [2]. Also, the motility, livability, and SCC of sperm were enhanced by Se treatments [46, 47].

As a result, we addressed the positive impacts on fresh semen quality of bucks treated with SeNPs enriched with SP combinations, but the main question 'Are bucks with these treatments, particularly during hot climates, have profits on sperm freezing and fertilizing capabilities after cryopreservation? The outcomes in this context showed that rabbit bucks with different levels of SeNP+SP combinations substantially improved sperm characteristics in post-thawed semen. This improvement concerns good quality semen produced by bucks in treatment groups. In rabbits, sperm characteristics in cryopreserved semen are associated with their characteristics before cryopreservation.

Antioxidant enzymes are reduced when bucks are exposed to HS [48]. Increasing MDA level causes organ damage in rabbits [49] in association with lower reproductive performance of bucks [1]. The harmful effects of HS on antioxidant enzymes and MDA levels were observed for bucks in the control-HS in comparison with the control-NC group. Therefore, decreasing oxidative stress is a crucial means of mitigating heat-stressed rabbits. The enhancement in the semen quality of bucks in treated groups in the current study was mainly related to the properties of either SP [17] or Se [2] alone and/or their synergetic effects, as antioxidants, which protect the cells from damage by enhancing antioxidant enzyme (GSH, GPx, and SOD) activity as well as reducing MDA level in serum and seminal plasma. The antioxidants scavenge ROS in the testes to decrease the endogenous oxidative damage by increasing antioxidant enzyme activities [50]. The antioxidant enzyme, SOD, converts superoxide anions to $H_2O_2$, which GPx catalyzes to $H_2O$ or alcohol [51]. The reduction in MDA production by SP and SeNPs combinations mitigated the oxidative stress condition. SP contains β-carotene, Se, polypeptides, pigments, tocopherol, and phenols. Phycocyanin, the primary pigment in SP, plays a vital role in potent antioxidant effects [52, 53]. In this context, both enzymatic and non-enzymatic antioxidant defense mechanisms were affected positively by Se addition in the diet of poultry [51].

It is worth noting that SeNPs enriched with SP at levels of 0.25 and 50 mg/kg diet improved the hematological blood parameters. A hematological and biochemical blood parameter

represents the animal's health [54]. Considering the findings of our study, SeNPs enriched with SP showed a significant increase in Hb, RBCs, PLT, and PCV compared to the control-HS group because the dietary photogenic, as an immune-modulatory agent, produced a significantly higher erythrogram than a control group in heat-stressed rabbits [23, 55, 56]. This improvement may indicate that treating SeNPs enriched with SP can protect rabbits from anemia under HS conditions [57].

Our results indicated deleterious effects of HS on all hematological parameters studied in terms of decreasing Hb concentration, RBCs and platelets count, and PCV (P<0.05) in control-Hs as compared to control-NC. It is worth noting that SeNPs enriched with SP at levels of 0.25 and 50 mg/kg diet improved the hematological blood parameters. A hematological and biochemical blood parameter represents the animal's health [54]. Considering the findings of our study, SeNPs enriched with SP showed a significant increase in Hb, RBCs, PLT, and PCV compared to the control-HS group because the dietary photogenic, as an immune-modulatory agent, produced a significantly higher erythrogram than a control group in heat-stressed rabbits [23, 55, 56]. This improvement may indicate that treating SeNPs enriched with SP can protect rabbits from anemia under HS conditions [57].

In our study, HS reduced total protein and increased TC concentrations in the serum of bucks in the control-HS, which agrees with the results of Liang et al. [9], who found a decrease in total protein and an increase in TC in heat-stressed rabbits. HS condition elevates the secretion of glucocorticoid, which stimulates gluconeogenesis [58]. The observed improvement of serum total protein and its main components (albumin and globulin) in association with reducing urea and creatinine levels in the serum of bucks in treatment groups compared to the control-HS may indicate improving protein utilization of treated bucks under HS. Also, the contents of proteins, essential amino acids, minerals, vitamins, phospholipids, and antioxidants in SP may cause improvement in protein utilization [15, 59]. The hypolipidemic activities of SP, SeNPs, or their combinations in reducing TC and TG, and increasing HDL concentration in serum were reported in animals and chickens [60, 61]. Dietary supplementation with a combination of SP and SeNPs for five weeks pre-mating reduced lipid peroxidation of heat-stressed rabbit does [31]. Supplemental SP with Zn (75 mg per kg diet) decreased total cholesterol and LDL levels in blood serum [12]. Such effect may be attributed to the fact that all supplements could decrease absorbed and synthesized cholesterol in the poultry gut [55]. In the same context, SP as an antioxidant contains polyphenolic contents, which may reduce the lipid level in the blood of treated rabbits by inhibiting pancreatic lipase activity and decreasing free fatty acid release into circulation [31, 62]. The results showed that AST and ALT activities increased in bucks exposed to HS in the control-HS. Similarly, Maria et al. [48] reported that HS showed damage and inflammation in the liver. According to the present results in our study, reducing AST and ALT activity as well as urea and creatinine in the serum of bucks in treatment groups may suggest a safe use of SP, SeNPs, or their combination on the normal liver and kidney functions of bucks.

In rabbits, semen variables are the significant factors affecting the reproductive performance of doe rabbits [7]. Rabbit bucks with strong sexual desire and high-quality semen are required throughout the year to achieve maximum productivity and libido through artificial insemination or natural mating [63]. SP+SeNPs combinations improved sexual desire, pregnancy rate, and fertility rate. Therefore, we can indicate that SP+SeNPs have significant positive impacts on the reproductive traits of doe rabbits [64]. Increasing the percentages of motility and normality, as well as concentrations of spermatozoa, is associated with an enhancement in fertility [1]. Additionally, sperm motility percentage affected the number of kits of doe rabbits. In mammals, sperm characteristics are essential in determining sperm fertility. The present study showed higher sperm fertility in a bucks-fed diet supplemented with

SP+SeNPs combinations due to the protection of spermatozoa from lipid peroxidation, which decrease the injury and damage of sperm cells, resulting in an enhancement in male fertility [2, 50, 65]. Overall, the rate of changes of the different studied variables indicates synergistic effects of SP-SeNPs, with higher results with increasing SeNPs from 25 to 50 mg in the mixture of SP with SeNPs.

## 5. Conclusion

Dietary inclusion of *spirulina platensis*, selenium nanoparticles, or their combinations (SP-SeNPs) could be potentially used as potent antioxidants for improving the reproductive performance of rabbit bucks by eliminating the deleterious effects of the oxidative stress under heat stress conditions via enhancing sexual desire, semen characteristics, fertilizing ability, health and antioxidant statuses, and immunity of rabbit bucks. The combinations of Spirulina (1 g) plus SeNPs at a level of 50 mg supplemented to the diet provides the most benefits, which can be recommended as a valuable tool for farming rabbit bucks utilized in natural mating or artificial insemination under heat stress circumstances.

## Supporting information

**S1 Data.**
(XLS)

## Author Contributions

**Conceptualization:** Ibrahim T. El-Ratel, Hamdy A. El-Nagar, Abdel-Khalek E. Abdel-Khalek, Mohammed Fouad El Basuini, Khaled H. El-Kholy, Sara F. Fouda.

**Data curation:** Ibrahim T. El-Ratel, Mawada E. Elbasuny, Hamdy A. El-Nagar, Abdel-Khalek E. Abdel-Khalek, Ali A. El-Raghi, Mohammed Fouad El Basuini, Khaled H. El-Kholy, Sara F. Fouda.

**Formal analysis:** Ibrahim T. El-Ratel, Mawada E. Elbasuny, Hamdy A. El-Nagar, Ali A. El-Raghi, Mohammed Fouad El Basuini, Khaled H. El-Kholy, Sara F. Fouda.

**Funding acquisition:** Ibrahim T. El-Ratel, Hamdy A. El-Nagar, Ali A. El-Raghi, Mohammed Fouad El Basuini, Khaled H. El-Kholy, Sara F. Fouda.

**Investigation:** Ibrahim T. El-Ratel, Mawada E. Elbasuny, Hamdy A. El-Nagar, Abdel-Khalek E. Abdel-Khalek, Ali A. El-Raghi, Khaled H. El-Kholy, Sara F. Fouda.

**Methodology:** Ibrahim T. El-Ratel, Mawada E. Elbasuny, Hamdy A. El-Nagar, Ali A. El-Raghi, Mohammed Fouad El Basuini, Khaled H. El-Kholy, Sara F. Fouda.

**Project administration:** Ibrahim T. El-Ratel, Hamdy A. El-Nagar, Abdel-Khalek E. Abdel-Khalek, Ali A. El-Raghi, Mohammed Fouad El Basuini, Khaled H. El-Kholy, Sara F. Fouda.

**Resources:** Ibrahim T. El-Ratel, Mawada E. Elbasuny, Hamdy A. El-Nagar, Abdel-Khalek E. Abdel-Khalek, Ali A. El-Raghi, Mohammed Fouad El Basuini, Sara F. Fouda.

**Software:** Ibrahim T. El-Ratel, Hamdy A. El-Nagar, Abdel-Khalek E. Abdel-Khalek, Mohammed Fouad El Basuini, Sara F. Fouda.

**Supervision:** Ibrahim T. El-Ratel, Hamdy A. El-Nagar, Abdel-Khalek E. Abdel-Khalek, Ali A. El-Raghi, Mohammed Fouad El Basuini, Khaled H. El-Kholy, Sara F. Fouda.

**Validation:** Ibrahim T. El-Ratel, Mawada E. Elbasuny, Hamdy A. El-Nagar, Abdel-Khalek E. Abdel-Khalek, Ali A. El-Raghi, Khaled H. El-Kholy, Sara F. Fouda.

**Visualization:** Ibrahim T. El-Ratel, Mawada E. Elbasuny, Hamdy A. El-Nagar, Abdel-Khalek E. Abdel-Khalek, Ali A. El-Raghi, Khaled H. El-Kholy, Sara F. Fouda.

**Writing – original draft:** Ibrahim T. El-Ratel, Mohammed Fouad El Basuini, Sara F. Fouda.

**Writing – review & editing:** Ibrahim T. El-Ratel, Mohammed Fouad El Basuini.

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
