## [Decision Letter · Decision Letter 0]

13 Feb 2023

PONE-D-23-02023The Synergistic Impact of Spirulina and Selenium Nanoparticles Mitigates the Adverse Effects of Heat Stress on Rabbits BucksPLOS ONE

Dear Dr. Mohammed Fouad El Basuini,

Thank you for submitting your manuscript to PLOS ONE. After careful consideration, we feel that it has merit but does not fully meet PLOS ONE’s publication criteria as it currently stands. Therefore, we invite you to submit a revised version of the manuscript that addresses the points raised during the review process.

Dear Dr., Mohammed Fouad El Basuini

Thank you for submitting your manuscript to PLOS ONE. After careful consideration, we feel that it has merit but does not fully meet PLOS ONE’s publication criteria as it currently stands. Therefore, we invite you to submit a revised version of the manuscript that addresses the points raised during the review process.

Thank you for submitting your manuscript to PLOS ONE. After careful consideration, we have decided that your manuscript needs Major Revision.

Kind regards,

Prof. Lamiaa Mostafa Radwan, Ph.D.

Academic Editor

PLOS ONE

Editor Comment

1-     Please authors do make data available.

2-     The title of the manuscript does not express the goal of the experiment well. It is suggested that the address (The Synergistic Impact of Spirulina and Selenium Nanoparticles Mitigates the Adverse Effects of Heat Stress on semen quality for Rabbits Bucks(.

3-     The abstract needs to write a one or two line at most explaining the economic importance of this research (the negative effect of heat stress on fertility and the decline in semen quality).

4-     The Material and methods need write in more detail (environment, managerial and hygienic conditions for rabbits during the experiment).

5-     Experimental design need write more clearly.  

6-     The results in general need more deep discussion.

7-     Clarification of the economic importance and the economic and environmental return of this experiment in all parts of the manuscript (summary, introduction, discussion of results and conclusion will strengthen the manuscript and clarify the goal of the experiment.

Reviewer1

The part of Materials and methods needs more clarification about design, were the all groups reared in the same building? cages type and dimensions, type of building, the period of the year you performed the experiment, have you treated all groups (except control) to heat stress treatment?.

Reviewer2

The manuscript represents a thorough and detailed research conducted by the researchers. Its highly commendable, though few grammatical errors were cited. If effected and published, it will stimulate readership. It also address a global challenge of mitigating adverse effects of global warming.

Reviewer3

Title needs modification.

It’s not clear what the authors wanted study;

a) Does they wanted to study effect of spirulina on HS and nano Selenium on HS alone or in combination?

b) If authors wish to study the interaction then why there is not combination supplementation in the positive control group (non-heat stress exposed group).

c) How the combination will be better than the individual ones?

The experimental design is not correct.

Introduction

The authors failed to highlight the need for study or impact of heat stress on the semen/reproductive performance in bucks.

Authors further failed to explain why spirulina and nano Se has been selected for the study? As there is various polyphenolic compounds and nano forms of other minerals (Zn, Cr); why these nutrients or additives has been not used in the study?

Why combination study is needed?

Materials and methods

What is the selenium content in the CPD?

What is the NRC recommended level of Selenium in rabbit diets?

What is the duration of supplementation or experiment period?

How semen physical quality was assessed? Was manual method or CASA was used?

The authors did not presented/exhibited the heat stress in bucks by exposing to cyclic HS.

No information about feed consumption since, the HS first affects the intake.

No data on feed consumption, body weight change or water intake as indicator of HS induction in bucks.

The results seems contradictors as HS could results in haemo-concentration then the blood parameters should be increased in negative control group compared to the TNZ or positive control. No discussion about the blood parameters were presented.

Why post-thaw semen quality was assessed? How HS is correlated with post-thaw semen quality?

Discussion is not clear and supplementing the findings. Needs major improvement.

The authors failed to explain the synergism between the spirulina and Se.

The manuscript needs major revision before considering for further processing.

We look forward to receiving your revised manuscript.

Kind regards,

Lamiaa Mostafa Radwan, Ph.D.

Academic Editor

PLOS ONE

Journal Requirements:

3. Please make sure that all information entered in the 'Ethics Statement' section regarding ethics approval is also included in the Methods section of the manuscript.

Additional Editor Comments:

Dear Dr., Mohammed Fouad El Basuini

Thank you for submitting your manuscript to PLOS ONE. After careful consideration, we feel that it has merit but does not fully meet PLOS ONE’s publication criteria as it currently stands. Therefore, we invite you to submit a revised version of the manuscript that addresses the points raised during the review process.

Thank you for submitting your manuscript to PLOS ONE. After careful consideration, we have decided that your manuscript needs Major Revision.

Kind regards,

Prof. Lamiaa Mostafa Radwan, Ph.D.

Academic Editor

PLOS ONE

Editor Comment

1- Please authors do make data available.

2- The title of the manuscript does not express the goal of the experiment well. It is suggested that the address (The Synergistic Impact of Spirulina and Selenium Nanoparticles Mitigates the Adverse Effects of Heat Stress on semen quality for Rabbits Bucks(.

3- The abstract needs to write a one or two line at most explaining the economic importance of this research (the negative effect of heat stress on fertility and the decline in semen quality).

4- The Material and methods need write in more detail (environment, managerial and hygienic conditions for rabbits during the experiment).

5- Experimental design need write more clearly.

6- The results in general need more deep discussion.

7- Clarification of the economic importance and the economic and environmental return of this experiment in all parts of the manuscript (summary, introduction, discussion of results and conclusion will strengthen the manuscript and clarify the goal of the experiment.

Reviewer1

The part of Materials and methods needs more clarification about design, were the all groups reared in the same building? cages type and dimensions, type of building, the period of the year you performed the experiment, have you treated all groups (except control) to heat stress treatment?.

Reviewer2

The manuscript represents a thorough and detailed research conducted by the researchers. Its highly commendable, though few grammatical errors were cited. If effected and published, it will stimulate readership. It also address a global challenge of mitigating adverse effects of global warming.

Reviewer3

Title needs modification.

It’s not clear what the authors wanted study;

a) Does they wanted to study effect of spirulina on HS and nano Selenium on HS alone or in combination?

b) If authors wish to study the interaction then why there is not combination supplementation in the positive control group (non-heat stress exposed group).

c) How the combination will be better than the individual ones?

The experimental design is not correct.

Introduction

The authors failed to highlight the need for study or impact of heat stress on the semen/reproductive performance in bucks.

Authors further failed to explain why spirulina and nano Se has been selected for the study? As there is various polyphenolic compounds and nano forms of other minerals (Zn, Cr); why these nutrients or additives has been not used in the study?

Why combination study is needed?

Materials and methods

What is the selenium content in the CPD?

What is the NRC recommended level of Selenium in rabbit diets?

What is the duration of supplementation or experiment period?

How semen physical quality was assessed? Was manual method or CASA was used?

The authors did not presented/exhibited the heat stress in bucks by exposing to cyclic HS.

No information about feed consumption since, the HS first affects the intake.

No data on feed consumption, body weight change or water intake as indicator of HS induction in bucks.

The results seems contradictors as HS could results in haemo-concentration then the blood parameters should be increased in negative control group compared to the TNZ or positive control. No discussion about the blood parameters were presented.

Why post-thaw semen quality was assessed? How HS is correlated with post-thaw semen quality?

Discussion is not clear and supplementing the findings. Needs major improvement.

The authors failed to explain the synergism between the spirulina and Se.

The manuscript needs major revision before considering for further processing.

Reviewers' comments:

Reviewer's Responses to Questions

**Comments to the Author**

1. Is the manuscript technically sound, and do the data support the conclusions?

Reviewer #1: Yes

Reviewer #2: Yes

Reviewer #3: Yes

2. Has the statistical analysis been performed appropriately and rigorously? 

Reviewer #1: Yes

Reviewer #2: Yes

Reviewer #3: Yes

3. Have the authors made all data underlying the findings in their manuscript fully available?

Reviewer #1: Yes

Reviewer #2: Yes

Reviewer #3: Yes

4. Is the manuscript presented in an intelligible fashion and written in standard English?

Reviewer #1: Yes

Reviewer #2: Yes

Reviewer #3: Yes

5. Review Comments to the Author

Reviewer #1: The part of Materials and methods needs more clarification about design, were the all groups reared in the same building? cages type and dimensions, type of building, the period of the year you performed the experiment, have you treated all groups (except control) to heat stress treatment?

Reviewer #2: The manuscript represents a thorough and detailed research conducted by the researchers. Its highly commendable, though few grammatical errors were cited. If effected and published, it will stimulate readership. It also address a global challenge of mitigating adverse effects of global warming.

Reviewer #3: find in the attachment.

Title needs modification.

It’s not clear what the authors wanted study;

a) Does they wanted to study effect of spirulina on HS and nano Selenium on HS alone or in combination?

b) If authors wish to study the interaction then why there is not combination supplementation in the positive control group (non-heat stress exposed group).

c) How the combination will be better than the individual ones?

The experimental design is not correct.

Introduction

The authors failed to highlight the need for study or impact of heat stress on the semen/reproductive performance in bucks.

Authors further failed to explain why spirulina and nano Se has been selected for the study? As there is various polyphenolic compounds and nano forms of other minerals (Zn, Cr); why these nutrients or additives has been not used in the study?

Why combination study is needed?

Materials and methods

What is the selenium content in the CPD?

What is the NRC recommended level of Selenium in rabbit diets?

What is the duration of supplementation or experiment period?

How semen physical quality was assessed? Was manual method or CASA was used?

The authors did not presented/exhibited the heat stress in bucks by exposing to cyclic HS.

No information about feed consumption since, the HS first affects the intake.

No data on feed consumption, body weight change or water intake as indicator of HS induction in bucks.

The results seems contradictors as HS could results in haemo-concentration then the blood parameters should be increased in negative control group compared to the TNZ or positive control. No discussion about the blood parameters were presented.

Why post-thaw semen quality was assessed? How HS is correlated with post-thaw semen quality?

Discussion is not clear and supplementing the findings. Needs major improvement.

The authors failed to explain the synergism between the spirulina and Se.

The manuscript needs major revision before considering for further processing.

6. PLOS authors have the option to publish the peer review history of their article (what does this mean?). If published, this will include your full peer review and any attached files.

Reviewer #1: No

Reviewer #2: **Yes: **Dr Jimoh Olatunji Abubakar

Reviewer #3: **Yes: **Marappan Gopi

---

## [Author Response · Author response to Decision Letter 0]

3 Mar 2023

Responses to the comments of the Editor

1- Please authors do make data available.

Response: Yes - all data are fully available without restriction.

2- The title of the manuscript does not express the goal of the experiment well. It is suggested that the address (The Synergistic Impact of Spirulina and Selenium Nanoparticles Mitigates the Adverse Effects of Heat Stress on semen quality for Rabbits Bucks).

Response: Ok, thanks a lot for this valuable comment. The title of the manuscript has been changed. Kindly check (line 2-3).

3- The abstract needs to write a one or two line at most explaining the economic importance of this research (the negative effect of heat stress on fertility and the decline in semen quality).

Response: We appreciate your suggestion and done accordingly. Kindly check (lines 25-30). The required modifications were made as follows:

“Heat stress has a detrimental effect on animal fertility, particularly testicular functions, including reduced sperm output and quality, which causes an economic loss in the production of rabbits. The present trial investigated the efficacy of dietary Spirulina (SP) (Arthrospira platensis), selenium nanoparticles (SeNPs), and their combination (SP-SeNPs) on semen quality, haemato-biochemical, oxidative stress, immunity, and sperm fertility of heat-stressed (HS) rabbit bucks.”

4- The Material and methods need write in more detail (environment, managerial and hygienic conditions for rabbits during the experiment).

Response: We deeply appreciate your comment. Kindly check (Lines 103-111 and Lines 119-126). The required modifications were made as follows:

“The study was conducted at a private commercial rabbit farm in Mansoura City, Dakahlia Governorate, Egypt, from December 2021 to April 2022. The experimental animals included 60 sexually mature APRI line bucks (7-8 months of age and 3.15±0.32 kg LBW). Each buck served as a replicate and was kept individually in stainless steel cage batteries (40×50×35cm) accommodated with feeders and automatic drinkers in a closed system with controlled heat, humidity, and light regimes (12 hrs light: 12 hrs dark) under the same managerial and hygienic conditions.”

“The experimental animals (n=60) were randomly divided into six groups (10/group). Bucks in the 1st group (control-NC, G1) were kept under normal conditions of ambient temperature (11-22 oC) and relative humidity (RH) from 40 to 45% and fed on CPD without supplementation. Bucks in the 2nd group (control-HS, G2) were exposed to HS conditions (32 ± 0.50 °C; 60-66% RH) and fed on CPD without supplementation. Other four treatment groups, including bucks kept under HS conditions as in G2 and fed on CPD supplemented with 1g SP (G3), 25 mg SeNPs (G4), 1 g SP+25 mg SeNPs (G5), and 1 g SP+50 mg SeNPs (G6) per kg diet, respectively. The treatment period was two months, followed by semen collection period of 10 weeks.”

5- Experimental design need write more clearly. 

Response: Thank you for your comment and done accordingly. Kindly check (line 119-126). 

6- The results in general need more deep discussion.

Response: Thanks a lot for this valuable comment, The results are discussed in more detail. Please check (lines 332-333, 338-339, 346-349, 405-414).

7- Clarification of the economic importance and the economic and environmental return of this experiment in all parts of the manuscript (summary, introduction, discussion of results and conclusion will strengthen the manuscript and clarify the goal of the experiment.

Response: Thanks a lot for this valuable comment, The results are discussed in more detail. Kindly check (summary line 25-27, introduction line 51-52 and discussion line 332-333, 338-339, 346-349, 405-414).

 

Responses to the comments of Reviewer #1

Comment: The part of Materials and methods needs more clarification about design, were the all groups reared in the same building? cages type and dimensions, type of building, the period of the year you performed the experiment, have you treated all groups (except control) to heat stress treatment?

Response: We sincerely appreciate your helpful and constructive comments. Kindly check (Lines 103-111 and Lines 119-126). The required modifications were made as follows:

Line 103-111:

“The study was conducted at a private commercial rabbit farm in Mansoura City, Dakahlia Governorate, Egypt, from December 2021 to April 2022. The experimental animals included 60 sexually mature APRI line bucks (7-8 months of age and 3.15±0.32 kg LBW). Each buck served as a replicate and was kept individually in stainless steel cage batteries (40×50×35cm) accommodated with feeders and automatic drinkers in a closed system with controlled heat, humidity, and light regimes (12 hrs light: 12 hrs dark) under the same managerial and hygienic conditions.”

Lines 119-126:

“The experimental animals (n=60) were randomly divided into six groups (10/group). Bucks in the 1st group (control-NC, G1) were kept under normal conditions of ambient temperature (11-22 oC) and relative humidity (RH) from 40 to 45% and fed on CPD without supplementation. Bucks in the 2nd group (control-HS, G2) were exposed to HS conditions (32 ± 0.50 °C; 60-66% RH) and fed on CPD without supplementation. Other four treatment groups, including bucks kept under HS conditions as in G2 and fed on CPD supplemented with 1g SP (G3), 25 mg SeNPs (G4), 1 g SP+25 mg SeNPs (G5), and 1 g SP+50 mg SeNPs (G6) per kg diet, respectively. The treatment period was two months, followed by semen collection period of 10 weeks.”

 

Responses to the comments of Reviewer #2

Comment: The manuscript represents a thorough and detailed research conducted by the researchers. Its highly commendable, though few grammatical errors were cited. If effected and published, it will stimulate readership. It also address a global challenge of mitigating adverse effects of global warming.

Response: we sincerely appreciate your effort in reviewing our manuscript and your constructive comments. The manuscript has been improved, organized, and proofread as suggested.

 

Responses to the comments of Reviewer #3

Comment: - Title needs modification. 

Response: Ok, thanks a lot for this valuable comment. The title of the manuscript has been changed. Kindly check (line 2-3).

Comment: - It’s not clear what the authors wanted study; a) Does they wanted to study effect of spirulina on HS and nano Selenium on HS alone or in combination? b) If authors wish to study the interaction then why there is not combination supplementation in the positive control group (non-heat stress exposed group). c) How the combination will be better than the individual ones? 

Response: Thank you very much for your constructive comment. Kindly check lines (27-30 and 119-126). The aim and design of the experiment has been clarified as follows:

Line 27-30:

“The present trial investigated the efficacy of dietary Spirulina (SP) (Arthrospira platensis), selenium nanoparticles (SeNPs), and their combination (SP-SeNPs) on semen quality, haemato-biochemical, oxidative stress, immunity, and sperm fertility of heat-stressed (HS) rabbit bucks.”

Lines 119-126:

“The experimental animals (n=60) were randomly divided into six groups (10/group). Bucks in the 1st group (control-NC, G1) were kept under normal conditions of ambient temperature (11-22 oC) and relative humidity (RH) from 40 to 45% and fed on CPD without supplementation. Bucks in the 2nd group (control-HS, G2) were exposed to HS conditions (32 ± 0.50 °C; 60-66% RH) and fed on CPD without supplementation. Other four treatment groups, including bucks kept under HS conditions as in G2 and fed on CPD supplemented with 1g SP (G3), 25 mg SeNPs (G4), 1 g SP+25 mg SeNPs (G5), and 1 g SP+50 mg SeNPs (G6) per kg diet, respectively. The treatment period was two months, followed by semen collection period of 10 weeks.”

Comment: The authors failed to highlight the need for study or impact of heat stress on the semen/reproductive performance in bucks. Authors further failed to explain why spirulina and nano Se has been selected for the study? As there is various polyphenolic compounds and nano forms of other minerals (Zn, Cr); why these nutrients or additives has been not used in the study? Why combination study is needed?

Response: Thank you for the comment. The adverse impacts of heat stress were detailed in lines 51-62. The importance of spirulina Se were written in lines 66-70. The use of SeNPs were detailed in lines 71-82. The combination studies (Spirulina + SeNPs) were in lines 86-93. Also, few studies were carried out on the effect of a combination of SP with different levels of SeNPs on the performance of male rabbits.

Comment: What is the selenium content in the CPD?

Response: Kindly check line 116.

Comment: What is the NRC recommended level of Selenium in rabbit diets?

Response: Thank you for the comment. Selenium levels in rabbit diets vary widely (0.05 - 1 mg/kg) according to dietary ingredients, rabbits' growth stage, and physiological condition.

Comment: What is the duration of supplementation or experiment period?

Response: Ok, thanks a lot for this valuable comment. The treatment period was two months followed by semen collection period of 10 weeks. Kindly check (line 126).

Comment: How semen physical quality was assessed? Was manual method or CASA was used?

Response: Thanks a lot for this valuable comment. All physical semen parameters were assessed manually.

Comment: The authors did not presented/exhibited the heat stress in bucks by exposing to cyclic HS.

Response: Thanks a lot for this valuable comment, the application of cyclic HS on semen production conflicts with the duration of spermatogenesis. So, we will need a recovery rate of about one month to study the effect of cyclic HS on sperm quality which was produced about one month before semen collection. In our study, we used a control group under normal conditions and another control group under HS conditions to assess the impaired effect of HS on different parameters studied.

Comment: No information about feed consumption since, the HS first affects the intake. No data on feed consumption, body weight change or water intake as indicator of HS induction in bucks.

Response: Thank you for your comment. Some data, such as feed consumption, body weight change or water intake, were assigned to a student graduation project and therefore were not included in the manuscript to avoid re-publishing the same data.

Comment: The results seems contradictors as HS could results in haemo-concentration then the blood parameters should be increased in negative control group compared to the TNZ or positive control. No discussion about the blood parameters were presented.

Response: Ok, thanks a lot for this valuable comment, blood parameters were presented in discussion. (Lines 401-410). 

Comment: Why post-thaw semen quality was assessed? How HS is correlated with post-thaw semen quality?

Response: Ok, thanks a lot for this valuable comment. The present results revealed the effect of treatments on semen quality in fresh cases. These results are good when we use the treated bucks in natural mating. The question is whether the semen produced from each group has the ability to cryopreserve or not. So, we studied the freezing ability of semen produced from each group when used in artificial insemination.

Comment: Discussion is not clear and supplementing the findings. Needs major improvement. The authors failed to explain the synergism between the spirulina and Se.

Response: We deeply appreciate your comment. The discussion part has been improved. Kindly check lines 332-333, 338-339, 346-349, 405-414.

---

## [Decision Letter · Decision Letter 1]

6 Apr 2023

PONE-D-23-02023R1The Synergistic Impact of Spirulina and Selenium Nanoparticles Mitigates the Adverse Effects of Heat Stress on Semen Quality of Rabbits BucksPLOS ONE

Dear Dr. El Basuini,

Thank you for submitting your manuscript to PLOS ONE. After careful consideration, we feel that it has merit but does not fully meet PLOS ONE’s publication criteria as it currently stands. Therefore, we invite you to submit a revised version of the manuscript that addresses the points raised during the review process.

Thank you for submitting your manuscript to PLOS ONE. After careful consideration, we have decided that your manuscript needs Major Revision.

Kind regards,

Prof. Lamiaa Mostafa Radwan, Ph.D.

Academic Editor

PLOS ONE

Editor Comment

I found your manuscript interesting and fit the journal scope. It has also improved well after the first revision round, but there are still some revisions that need to be fixed before accepting this paper.

Please make the modifications requested by the reviewers shown below

**Reviewer 4**

Dear Authors,

I found your MS interesting and fit the journal scope. It has also improved well after the first revision round, but there are still some revisions need to be fixed before accepting this paper.

Introduction:

There are some important recent related references that can be added as a background to support your introduction such as:

- Effect of selenium nanoparticles and/or Spirulina Platensis on growth, hematobiochemical, antioxidant

status, hormonal profile, immunity, and apoptosis of growing rabbits exposed to thermal stress.

- Impacts of selenium nanoparticles and spirulina alga to alleviate the deleterious effects of heat stress on reproductive efficiency, oxidative capacity, and immunity of doe rabbits.

- Zinc and/or Selenium enriched Spirulina as antioxidants in growing rabbit diets to alleviate the deleterious impacts of heat stress during summer season.

- Effect of rearing system and season on behavior, productive performance and carcass quality of rabbit: a review.

L.49: “[2-5]”: How you started your references' sequence with number 2!

L.58: Change “Also” to “Additionally”.

L.60: “ROS”: Define it (expand) as it is mentioned for the first time in your text.

L.63: “8–11”: Please, add the following reference to support your statement:

- Effect of bee venom on reproductive performance and immune response of male rabbits.

L.72: “so”: Add “therefore,” instead.

L.410-414: Try to go deeper in the scientific explanation. You can also refer to my suggested reference titles mentioned for the introduction to enrich your discussion as a general.

L.416: “[7]”: Please, add the following related references to support your current statement and your conclusion (as you mentioned "artificial insemination under heat stress"):

- Soliman, F. and El-Sabrout, K. (2020). Artificial insemination in rabbits: Factors that interfere in assessing its results. Journal of Animal Behaviour and Biometeorology, 8, 120–130. https://doi.org/10.31893/jabb.20016

- Elkomy, A.; El-Hanoun, A.; Abdella, M.; El-Sabrout, K. Improving the reproductive, immunity and health status of rabbit does using honey bee venom. J. Anim. Physiol. Anim. Nutr. 2021, 105, 975–983. https://doi.org/10.1111/jpn.13552

**Reviewer 5**

Manuscript title: The Synergistic Impact of Spirulina and Selenium Nanoparticles Mitigates the Adverse Effects of Heat Stress on Rabbits Bucks

The manuscript addresses a novel and vital topic for researchers, breeders, and others interested in the field. However, the following comments, inquiries, and suggestions may help improve the manuscript's quality.

1- How do you judge that the conditions in the positive group are thermal-neutral?

2- The hypothesis is not clear in the introduction. (See the comment (P:2) of the comments L:82-83).

3- I recommend calculating the change caused by the combination of Spirulina (SP) and two levels of selenium nanoparticles (SP-SeNPs) in a new table.

- Values that have already exceeded the average value = SP value + SP-SeNPs value/2 are placed. Keeping in mind that only the superior values are placed in the new table, and the values that are placed have a positive meaning, regardless of whether the values are + or -.

- Reference should be made to the new table which you will do with the results and discussion to clarify the meaning of the synergy between the combination of SP and SP-SeNPs.

- Indicate which of the two levels is the best or both according to the results in the new table and refer to this in the abstract and in the conclusion

- To obtain the change % in the following

1- Calculating the average value of the (SP + SP-SeNPs).

2- Calculating the value of the change in the value when the combination of SP and SP-SeNPs in both the two levels;

The change = recorded value - (1).

3- Calculate the change as a percentage;

Change % = (2) / (1) X 100.

Line:28, 112 ……in the 1st group

Comment: a superscript letter

Line 57-58:…..

Comment: It is necessary to specify the type of change that occurred, whether by increase or decrease, rather than writing a change in general.

In addition to I attached the comments

We look forward to receiving your revised manuscript.

Kind regards,

Lamiaa Mostafa Radwan, Ph.D.

Academic Editor

PLOS ONE

Additional Editor Comments:

Dear Dr. El Basuini,

Thank you for submitting your manuscript to PLOS ONE. After careful consideration, we feel that it has merit but does not fully meet PLOS ONE’s publication criteria as it currently stands. Therefore, we invite you to submit a revised version of the manuscript that addresses the points raised during the review process.

Thank you for submitting your manuscript to PLOS ONE. After careful consideration, we have decided that your manuscript needs Major Revision.

Kind regards,

Prof. Lamiaa Mostafa Radwan, Ph.D.

Academic Editor

PLOS ONE

Editor Comment

I found your manuscript interesting and fit the journal scope. It has also improved well after the first revision round, but there are still some revisions that need to be fixed before accepting this paper.

Please make the modifications requested by the reviewers shown below

Reviewer 4

Dear Authors,

I found your MS interesting and fit the journal scope. It has also improved well after the first revision round, but there are still some revisions need to be fixed before accepting this paper.

Introduction:

There are some important recent related references that can be added as a background to support your introduction such as:

- Effect of selenium nanoparticles and/or Spirulina Platensis on growth, hematobiochemical, antioxidant

status, hormonal profile, immunity, and apoptosis of growing rabbits exposed to thermal stress.

- Impacts of selenium nanoparticles and spirulina alga to alleviate the deleterious effects of heat stress on reproductive efficiency, oxidative capacity, and immunity of doe rabbits.

- Zinc and/or Selenium enriched Spirulina as antioxidants in growing rabbit diets to alleviate the deleterious impacts of heat stress during summer season.

- Effect of rearing system and season on behavior, productive performance and carcass quality of rabbit: a review.

L.49: “[2-5]”: How you started your references' sequence with number 2!

L.58: Change “Also” to “Additionally”.

L.60: “ROS”: Define it (expand) as it is mentioned for the first time in your text.

L.63: “8–11”: Please, add the following reference to support your statement:

- Effect of bee venom on reproductive performance and immune response of male rabbits.

L.72: “so”: Add “therefore,” instead.

L.410-414: Try to go deeper in the scientific explanation. You can also refer to my suggested reference titles mentioned for the introduction to enrich your discussion as a general.

L.416: “[7]”: Please, add the following related references to support your current statement and your conclusion (as you mentioned "artificial insemination under heat stress"):

- Soliman, F. and El-Sabrout, K. (2020). Artificial insemination in rabbits: Factors that interfere in assessing its results. Journal of Animal Behaviour and Biometeorology, 8, 120–130. https://doi.org/10.31893/jabb.20016

- Elkomy, A.; El-Hanoun, A.; Abdella, M.; El-Sabrout, K. Improving the reproductive, immunity and health status of rabbit does using honey bee venom. J. Anim. Physiol. Anim. Nutr. 2021, 105, 975–983. https://doi.org/10.1111/jpn.13552

Reviewer 5

Manuscript title: The Synergistic Impact of Spirulina and Selenium Nanoparticles Mitigates the Adverse Effects of Heat Stress on Rabbits Bucks

The manuscript addresses a novel and vital topic for researchers, breeders, and others interested in the field. However, the following comments, inquiries, and suggestions may help improve the manuscript's quality.

1- How do you judge that the conditions in the positive group are thermal-neutral?

2- The hypothesis is not clear in the introduction. (See the comment (P:2) of the comments L:82-83).

3- I recommend calculating the change caused by the combination of Spirulina (SP) and two levels of selenium nanoparticles (SP-SeNPs) in a new table.

- Values that have already exceeded the average value = SP value + SP-SeNPs value/2 are placed. Keeping in mind that only the superior values are placed in the new table, and the values that are placed have a positive meaning, regardless of whether the values are + or -.

- Reference should be made to the new table which you will do with the results and discussion to clarify the meaning of the synergy between the combination of SP and SP-SeNPs.

- Indicate which of the two levels is the best or both according to the results in the new table and refer to this in the abstract and in the conclusion

- To obtain the change % in the following

1- Calculating the average value of the (SP + SP-SeNPs).

2- Calculating the value of the change in the value when the combination of SP and SP-SeNPs in both the two levels;

The change = recorded value - (1).

3- Calculate the change as a percentage;

Change % = (2) / (1) X 100.

Line:28, 112 ……in the 1st group

Comment: a superscript letter

Line 57-58:…..

Comment: It is necessary to specify the type of change that occurred, whether by increase or decrease, rather than writing a change in general.

In addition to I attached the comments

Reviewers' comments:

Reviewer's Responses to Questions

**Comments to the Author**

1. If the authors have adequately addressed your comments raised in a previous round of review and you feel that this manuscript is now acceptable for publication, you may indicate that here to bypass the “Comments to the Author” section, enter your conflict of interest statement in the “Confidential to Editor” section, and submit your "Accept" recommendation.

Reviewer #4: (No Response)

Reviewer #5: All comments have been addressed

2. Is the manuscript technically sound, and do the data support the conclusions?

Reviewer #4: Yes

Reviewer #5: Yes

3. Has the statistical analysis been performed appropriately and rigorously? 

Reviewer #4: (No Response)

Reviewer #5: Yes

4. Have the authors made all data underlying the findings in their manuscript fully available?

Reviewer #4: (No Response)

Reviewer #5: Yes

5. Is the manuscript presented in an intelligible fashion and written in standard English?

Reviewer #4: (No Response)

Reviewer #5: Yes

6. Review Comments to the Author

Reviewer #4: Dear Authors,

I found your MS interesting and fit the journal scope. It has also improved well after the first revision round, but there are still some revisions need to be fixed before accepting this paper.

Introduction:

There are some important recent related references that can be added as a background to support your introduction such as:

- Effect of selenium nanoparticles and/or Spirulina Platensis on growth, hematobiochemical, antioxidant

status, hormonal profile, immunity, and apoptosis of growing rabbits exposed to thermal stress.

- Impacts of selenium nanoparticles and spirulina alga to alleviate the deleterious effects of heat stress on reproductive efficiency, oxidative capacity, and immunity of doe rabbits.

- Zinc and/or Selenium enriched Spirulina as antioxidants in growing rabbit diets to alleviate the deleterious impacts of heat stress during summer season.

- Effect of rearing system and season on behavior, productive performance and carcass quality of rabbit: a review.

L.49: “[2-5]”: How you started your references' sequence with number 2!

L.58: Change “Also” to “Additionally”.

L.60: “ROS”: Define it (expand) as it is mentioned for the first time in your text.

L.63: “8–11”: Please, add the following reference to support your statement:

- Effect of bee venom on reproductive performance and immune response of male rabbits.

L.72: “so”: Add “therefore,” instead.

L.410-414: Try to go deeper in the scientific explanation. You can also refer to my suggested reference titles mentioned for the introduction to enrich your discussion as a general.

L.416: “[7]”: Please, add the following related references to support your current statement and your conclusion (as you mentioned "artificial insemination under heat stress"):

- Soliman, F. and El-Sabrout, K. (2020). Artificial insemination in rabbits: Factors that interfere in assessing its results. Journal of Animal Behaviour and Biometeorology, 8, 120–130. https://doi.org/10.31893/jabb.20016

- Elkomy, A.; El-Hanoun, A.; Abdella, M.; El-Sabrout, K. Improving the reproductive, immunity and health status of rabbit does using honey bee venom. J. Anim. Physiol. Anim. Nutr. 2021, 105, 975–983. https://doi.org/10.1111/jpn.13552

Reviewer #5: Manuscript title: The Synergistic Impact of Spirulina and Selenium Nanoparticles Mitigates the Adverse Effects of Heat Stress on Rabbits Bucks

The manuscript addresses a novel and vital topic for researchers, breeders, and others interested in the field. However, the following comments, inquiries, and suggestions may help improve the manuscript's quality.

1- How do you judge that the conditions in the positive group are thermal-neutral?

2- The hypothesis is not clear in the introduction. (See the comment (P:2) of the comments L:82-83).

3- I recommend calculating the change caused by the combination of Spirulina (SP) and two levels of selenium nanoparticles (SP-SeNPs) in a new table.

- Values that have already exceeded the average value = SP value + SP-SeNPs value/2 are placed. Keeping in mind that only the superior values are placed in the new table, and the values that are placed have a positive meaning, regardless of whether the values are + or -.

- Reference should be made to the new table which you will do with the results and discussion to clarify the meaning of the synergy between the combination of SP and SP-SeNPs.

- Indicate which of the two levels is the best or both according to the results in the new table and refer to this in the abstract and in the conclusion

- To obtain the change % in the following

1- Calculating the average value of the (SP + SP-SeNPs).

2- Calculating the value of the change in the value when the combination of SP and SP-SeNPs in both the two levels;

The change = recorded value - (1).

3- Calculate the change as a percentage;

Change % = (2) / (1) X 100.

Line:28, 112 ……in the 1st group

Comment: a superscript letter

Line 57-58:…..

Comment: It is necessary to specify the type of change that occurred, whether by increase or decrease, rather than writing a change in general.

In addition to I attached the comments

7. PLOS authors have the option to publish the peer review history of their article (what does this mean?). If published, this will include your full peer review and any attached files.

Reviewer #4: No

Reviewer #5: **Yes: **Abdelazeem S. Abdelazeem

---

## [Author Response · Author response to Decision Letter 1]

9 Apr 2023

Responses to the comments of the Editor

I found your manuscript interesting and fit the journal scope. It has also improved well after the first revision round, but there are still some revisions that need to be fixed before accepting this paper. Please make the modifications requested by the reviewers shown below.

Response:

Thank you for your time and consideration. Your comments and those of the reviewers were highly insightful and enabled us to enhance the quality of our manuscript. We have now changed this manuscript and inserted the required modifications with point-by-point responses to each of the reviewer’s comments. The corrected sentences were followed using Track Changes in the revised manuscript. 

Responses to the comments of Reviewer #4

Dear Authors, I found your MS interesting and fit the journal scope. It has also improved well after the first revision round, but there are still some revisions need to be fixed before accepting this paper.

Response: We sincerely appreciate your helpful and constructive comments. The manuscript has been improved as suggested.

Introduction: There are some important recent related references that can be added as a background to support your introduction such as:

- Effect of selenium nanoparticles and/or Spirulina Platensis on growth, hematobiochemical, antioxidant status, hormonal profile, immunity, and apoptosis of growing rabbits exposed to thermal stress.

- Impacts of selenium nanoparticles and spirulina alga to alleviate the deleterious effects of heat stress on reproductive efficiency, oxidative capacity, and immunity of doe rabbits.

- Zinc and/or Selenium enriched Spirulina as antioxidants in growing rabbit diets to alleviate the deleterious impacts of heat stress during summer season.

- Effect of rearing system and season on behavior, productive performance and carcass quality of rabbit: a review.

Response: We appreciate your suggestion and done accordingly. Kindly check (lines 73, 97, 99).

L.49: “[2-5]”: How you started your references' sequence with number 2!

Response: Thank you for the comment. Citations sequels were corrected. Kindly check (line 56).

L.58: Change “Also” to “Additionally”.

Response: Thank you for your comment and done accordingly. Kindly check (line 65).

L.60: “ROS”: Define it (expand) as it is mentioned for the first time in your text.

Response: Ok, thanks a lot for this valuable comment. ROS has been defined as reactive oxygen species (ROS). Please check (Line 71).

L.63: “8–11”: Please, add the following reference to support your statement: - Effect of bee venom on reproductive performance and immune response of male rabbits.

Response: We appreciate your suggestion and done accordingly. Kindly check (line 73).

L.72: “so”: Add “therefore,” instead.

Response: Done as suggested. Kindly check (line 82).

L.410-414: Try to go deeper in the scientific explanation. You can also refer to my suggested reference titles mentioned for the introduction to enrich your discussion as a general.

Response: We appreciate your suggestion and done accordingly. Kindly check (lines 448-455).

L.416: “[7]”: Please, add the following related references to support your current statement and your conclusion (as you mentioned "artificial insemination under heat stress"):

- Soliman, F. and El-Sabrout, K. (2020). Artificial insemination in rabbits: Factors that interfere in assessing its results. Journal of Animal Behaviour and Biometeorology, 8, 120–130. https://doi.org/10.31893/jabb.20016

- Elkomy, A.; El-Hanoun, A.; Abdella, M.; El-Sabrout, K. Improving the reproductive, immunity and health status of rabbit does using honey bee venom. J. Anim. Physiol. Anim. Nutr. 2021, 105, 975–983. https://doi.org/10.1111/jpn.13552

Response: We appreciate your suggestion and done accordingly. Kindly check (lines 466-468).

Responses to the comments of Reviewer #5

Manuscript title: The Synergistic Impact of Spirulina and Selenium Nanoparticles Mitigates the Adverse Effects of Heat Stress on Rabbits Bucks

The manuscript addresses a novel and vital topic for researchers, breeders, and others interested in the field. However, the following comments, inquiries, and suggestions may help improve the manuscript's quality.

Response: We deeply appreciate your helpful and constructive comments. The manuscript has been improved as suggested.

1. How do you judge that the conditions in the positive group are thermal-neutral?

Response: Thank you for the comment. Bucks in the 1st group (control-NC) were kept under normal conditions (11-22 oC; 40 - 45% RH%), while the other groups were kept under heat stress conditions (32±0.50 °C; 60-66% RH %). Kindly check (lines 29-32 and 135-138)

2. The hypothesis is not clear in the introduction. (See the comment (P:2) of the comments L:82-83).

Response: Ok, we appreciate your suggestion and was done accordingly. Kindly check (lines 102-103)

3. I recommend calculating the change caused by the combination of Spirulina (SP) and two levels of selenium nanoparticles (SP-SeNPs) in a new table. 

- Values that have already exceeded the average value = SP value + SP-SeNPs value/2 are placed. Keeping in mind that only the superior values are placed in the new table, and the values that are placed have a positive meaning, regardless of whether the values are + or -.

- Reference should be made to the new table which you will do with the results and discussion to clarify the meaning of the synergy between the combination of SP and SP-SeNPs. 

- Indicate which of the two levels is the best or both according to the results in the new table and refer to this in the abstract and in the conclusion

- To obtain the change % in the following 

1- Calculating the average value of the (SP + SP-SeNPs).

2- Calculating the value of the change in the value when the combination of SP and SP-SeNPs in both the two levels;

The change = recorded value - (1).

3- Calculate the change as a percentage; 

Change % = (2) / (1) X 100.

Table 8. The percentage change in both levels due to the synergistic impact of the SP and SP-SeNPs.

Items Average Change %

SP+SeNPs25 Change %

SP+SeNPs50

Hematological variables

Hemoglobin (11.93+ 10.5) /2

= 11.215 (12.19-11.215) /11.215*100

= 8.69 (12.28-11.215) /11.215*100

= 9.5

RBCs 6.18 2.75 2.91

…ect 

Blood metabolites

…..ect 

antioxidant and oxidative marks in the blood and seminal plasma

…. Ect 

Semen production parameters

….. 

Sperm variables in post-thawed

…. Ect 

Blood and seminal testosterone

…..ect 

Reproductive performance of rabbits does

……ect 

Response: Ok, we appreciate your suggestion and was done accordingly. Kindly check Table 8 and Lines 337-350.

Line:28, 112 ……in the 1st group Comment: a superscript letter

Response: Done as requested. Kindly check line 29.

Line 57-58: Comment: It is necessary to specify the type of change that occurred, whether by increase or decrease, rather than writing a change in general.

Response: We deeply appreciate your constructive comment. The types of changes were specified in Lines 65-67.

Line 59, 328, 369. Comment: Put the whole term in the first time in L: 59, and the abbreviation in the next. The term ROS = reactive oxygen species.

Response: Done as requested. Kindly check line 71.

Line 63………..and antiparasitic activities and contains. Comment: Remove the spaces.

Response: Done as requested. Kindly check lines 74-75.

Line 67……….. animal nuitrition, and the most Comment: animal nutrition and the most.

Response: Corrected as requested. Kindly check line 79.

Line 70 . So, elements in NPs form…Comment: So, elements in the NPs form

Response: Corrected as requested. Kindly check line 82.

Line 73-74…, metabolic processes, and is an essential trace element for animals. Comment:….., and metabolic processes, and is an essential trace element for animals.

Response: Done as requested. Kindly check line 86.

Line 74….Se is a crucial component in …….Comment:… The Se is a crucial component in

Response: Corrected as requested. Kindly check line 87.

Line 82-83: SeNPs and/or Spirulina (SP) were incorporated to mitigate the negative impacts on the heat-stressed growing rabbit by improving the antioxidant status. Comment: Rewrite this sentence to be the research hypothesis.

Response: Done as requested. Kindly check lines 101-103.

Line 97………and 3.15±0.32 kg LBW Comment: don't use the abbreviation (LBW) and use the full term "Live body weight".

Response: Done as requested. Kindly check line 120.

Line 97-99…..Each buck served as a replicate and was kept individually in stainless steel cage batteries (40×50×35cm) accommodated with feeders and automatic drinkers in a closed system controlled by heat, humidity, and light regimes. Comment: Please remove the spaces in the previous sentence.

Response: Done as requested. Kindly check lines 121-124.

Line 113 … under normal indoor conditions. Comment : write the (…-… °C; …-…RH%).

Response: Done as requested. Kindly check lines 135-136.

Line 116 - 117….Comment : What was the basis for selecting the number of hours and days that you relied on in the negative control group?

Response: We deeply appreciate your constructive comment. The basis of selection was based on a previous study (Please check line 138:

El-Ratel IT, Attia KAH, El-Raghi AA, Fouda SF. Relief of the negative effects of heat stress on semen quality, reproductive efficiency and oxidative capacity of rabbit bucks using different natural antioxidants. Anim Biosci. 2021;34: 844–854. doi:10.5713/ajas.20.0258 

Line 125…. The mean PDI was 0.13, Comment : Don't use the abbreviation (PDI) and use the full term Polydispersity Index.

Response: Done as requested. Kindly check line 147.

Line 143…. by using a phase-contrast microscope with a hot stage at 37°C 

Comment: Remove the spaces.

Response: Done as requested. Kindly check line 165.

Line 161 Comment: Tab.

Response: Done as requested. Kindly check line 183.

Line 176 ….. ear-vein of the does Comment: ear-vein of the dose

Response: Corrected as suggested. Kindly check line 199.

Line 268: Table 3.. .. and their combinations on oxidative bio-marks in blood and...Comment:….., and their combinations on antioxidant and oxidative marks in the blood and …. 

Response: Corrected as requested. Kindly check line 290.

---

## [Decision Letter · Decision Letter 2]

24 Apr 2023

PONE-D-23-02023R2The Synergistic Impact of Spirulina and Selenium Nanoparticles Mitigates the Adverse Effects of Heat Stress on Semen Quality of Rabbits BucksPLOS ONE

Dear Dr.  El Basuini,

Thank you for submitting your manuscript to PLOS ONE. After careful consideration, we feel that it has merit but does not fully meet PLOS ONE’s publication criteria as it currently stands. Therefore, we invite you to submit a revised version of the manuscript that addresses the points raised during the review process.

Dear Dr., Mohammed Fouad El Basuini

Thank you for submitting your manuscript to PLOS ONE. After careful consideration, we feel that it has merit but does not fully meet PLOS ONE’s publication criteria as it currently stands. Therefore, we invite you to submit a revised version of the manuscript that addresses the points raised during the review process.

Thank you for submitting your manuscript to PLOS ONE. After careful consideration, we have decided that your manuscript needs Minor Revision.

Kind regards,

Prof. Lamiaa Mostafa Radwan, Ph.D.

Academic Editor

PLOS ONE

Editor Comment

I found your manuscript interesting and fit the journal's scope. It has also improved well after the second revision round, but there are still some revisions that need to be fixed before accepting this paper.

Please make the modifications requested by the reviewers shown below (Reviewers, 2 and 6)

**Reviewer 1**

Regarding the manuscript entitled "The Synergistic Impact of Spirulina and Selenium Nanoparticles Mitigates the Adverse

Effects of Heat Stress on Semen Quality of Rabbits Bucks"

Abstract: Heat stress has a detrimental effect on animal fertility, particularly testicular functions,

including reduced sperm output and quality, which causes an economic loss in the

production of rabbits. The present trial investigated the efficacy of dietary Spirulina

(SP) (Arthrospira platensis), selenium nanoparticles (SeNPs), and their combination

(SP-SeNPs) on semen quality, haemato-biochemical, oxidative stress, immunity, and

sperm quality of heat-stressed (HS) rabbit bucks. Sixty mature bucks (APRI line) were

distributed into 6 groups of ten replicates under controlled conditions. Bucks in the 1st

group (control-NC) were kept under normal conditions (11-22 oC; 40 - 45% RH% =

relative humidity), while the 2nd group (control-HS) was kept under heat stress

conditions (32±0.50 °C; 60-66% RH %). The control groups were fed a commercial

pelleted diet and the other four heat-stressed groups were fed a commercial pelleted

diet with 1 g SP, 25 mg SeNPs, 1 g SP+25 mg SeNPs, and 1 g SP+50 mg SeNPs per

kg diet, respectively. The dietary inclusion of SP, SeNPs, and their combinations

significantly increased hemoglobin, platelets, total serum protein, high-density

lipoproteins, glutathione, glutathione peroxidase, superoxide dismutase, and seminal

plasma testosterone while decreased triglycerides, total cholesterol, urea, creatinine,

and malondialdehyde compared with the control-HS. Red blood cells, packed cell

volume, serum albumin, and testosterone significantly increased, while SeNPs,

SP+SeNPs25, and SP+SeNPs50 significantly decreased low-density lipoproteins,

aspartate, and alanine amino transferees. Total antioxidant capacity substantially

increased in serum and seminal plasma, while seminal plasma malondialdehyde

decreased in 25 or 50 mg of SeNPs+SP/kg groups. All supplements significantly

improved libido, sperm livability, concentration, intact acrosome, membrane integrity,

total output in fresh semen, and sperm quality in cryopreserved semen. SP-SeNPs50

had higher synergistic effect than SP-SeNPs25 on most different variables studied. In

conclusion, the dietary inclusion of SP plus SeNPs50 has a synergistic effect and is

considered a suitable dietary supplement for improving reproductive efficiency, health,

oxidative stress, and immunity of bucks in the breeding strategy under hot climates.

all comments have been addressed in the revised version nu authors.

**Reviewer 2**

All comments have been addressed

Please look the attached file

**Reviewer 4**

Dear Authors,

I am pleased to inform you that I accepted your manuscript in the current form.

Regards

**Reviewer 5**

All of my suggestions were incorporated into the revised manuscript.

**Reviewer 6**

The following corrections need to be made:

Line 74….Se is a crucial component in …….Comment:… The Se is a crucial component in Response: Corrected as requested. Kindly check line 87.

My comment: Above correction was not done

Line 82-83: SeNPs and/or Spirulina (SP) were incorporated to mitigate the negative impacts on the heat-stressed growing rabbit by improving the antioxidant status. Comment: Rewrite this sentence to be the research hypothesis. Response: Done as requested. Kindly check lines 101-103.

My comment: L 98-99: The phrase “may affect the performance of male rabbits” is too general. Will the effect be positive or negative? What kind of performance in male rabbits will be affected?

We look forward to receiving your revised manuscript.

Kind regards,

Lamiaa Mostafa Radwan, Ph.D.

Academic Editor

PLOS ONE

Journal Requirements:

Additional Editor Comments:

Dear Dr., Mohammed Fouad El Basuini

Thank you for submitting your manuscript to PLOS ONE. After careful consideration, we feel that it has merit but does not fully meet PLOS ONE’s publication criteria as it currently stands. Therefore, we invite you to submit a revised version of the manuscript that addresses the points raised during the review process.

Thank you for submitting your manuscript to PLOS ONE. After careful consideration, we have decided that your manuscript needs Minor Revision.

Kind regards,

Prof. Lamiaa Mostafa Radwan, Ph.D.

Academic Editor

PLOS ONE

Editor Comment

I found your manuscript interesting and fit the journal's scope. It has also improved well after the second revision round, but there are still some revisions that need to be fixed before accepting this paper.

Please make the modifications requested by the reviewers shown below (Reviewers, 2 and 6)

Reviewer 1

Regarding the manuscript entitled "The Synergistic Impact of Spirulina and Selenium Nanoparticles Mitigates the Adverse

Effects of Heat Stress on Semen Quality of Rabbits Bucks"

Abstract: Heat stress has a detrimental effect on animal fertility, particularly testicular functions,

including reduced sperm output and quality, which causes an economic loss in the

production of rabbits. The present trial investigated the efficacy of dietary Spirulina

(SP) (Arthrospira platensis), selenium nanoparticles (SeNPs), and their combination

(SP-SeNPs) on semen quality, haemato-biochemical, oxidative stress, immunity, and

sperm quality of heat-stressed (HS) rabbit bucks. Sixty mature bucks (APRI line) were

distributed into 6 groups of ten replicates under controlled conditions. Bucks in the 1st

group (control-NC) were kept under normal conditions (11-22 oC; 40 - 45% RH% =

relative humidity), while the 2nd group (control-HS) was kept under heat stress

conditions (32±0.50 °C; 60-66% RH %). The control groups were fed a commercial

pelleted diet and the other four heat-stressed groups were fed a commercial pelleted

diet with 1 g SP, 25 mg SeNPs, 1 g SP+25 mg SeNPs, and 1 g SP+50 mg SeNPs per

kg diet, respectively. The dietary inclusion of SP, SeNPs, and their combinations

significantly increased hemoglobin, platelets, total serum protein, high-density

lipoproteins, glutathione, glutathione peroxidase, superoxide dismutase, and seminal

plasma testosterone while decreased triglycerides, total cholesterol, urea, creatinine,

and malondialdehyde compared with the control-HS. Red blood cells, packed cell

volume, serum albumin, and testosterone significantly increased, while SeNPs,

SP+SeNPs25, and SP+SeNPs50 significantly decreased low-density lipoproteins,

aspartate, and alanine amino transferees. Total antioxidant capacity substantially

increased in serum and seminal plasma, while seminal plasma malondialdehyde

decreased in 25 or 50 mg of SeNPs+SP/kg groups. All supplements significantly

improved libido, sperm livability, concentration, intact acrosome, membrane integrity,

total output in fresh semen, and sperm quality in cryopreserved semen. SP-SeNPs50

had higher synergistic effect than SP-SeNPs25 on most different variables studied. In

conclusion, the dietary inclusion of SP plus SeNPs50 has a synergistic effect and is

considered a suitable dietary supplement for improving reproductive efficiency, health,

oxidative stress, and immunity of bucks in the breeding strategy under hot climates.

all comments have been addressed in the revised version nu authors.

Reviewer 2

All comments have been addressed

Please look the attached file

Reviewer 4

Dear Authors,

I am pleased to inform you that I accepted your manuscript in the current form.

Regards

Reviewer 5

All of my suggestions were incorporated into the revised manuscript.

Reviewer 6

The following corrections need to be made:

Line 74….Se is a crucial component in …….Comment:… The Se is a crucial component in Response: Corrected as requested. Kindly check line 87.

My comment: Above correction was not done

Line 82-83: SeNPs and/or Spirulina (SP) were incorporated to mitigate the negative impacts on the heat-stressed growing rabbit by improving the antioxidant status. Comment: Rewrite this sentence to be the research hypothesis. Response: Done as requested. Kindly check lines 101-103.

My comment: L 98-99: The phrase “may affect the performance of male rabbits” is too general. Will the effect be positive or negative? What kind of performance in male rabbits will be affected?

Reviewers' comments:

Reviewer's Responses to Questions

**Comments to the Author**

1. If the authors have adequately addressed your comments raised in a previous round of review and you feel that this manuscript is now acceptable for publication, you may indicate that here to bypass the “Comments to the Author” section, enter your conflict of interest statement in the “Confidential to Editor” section, and submit your "Accept" recommendation.

Reviewer #1: All comments have been addressed

Reviewer #2: All comments have been addressed

Reviewer #4: (No Response)

Reviewer #5: All comments have been addressed

Reviewer #6: (No Response)

2. Is the manuscript technically sound, and do the data support the conclusions?

Reviewer #1: Yes

Reviewer #2: Yes

Reviewer #4: (No Response)

Reviewer #5: Yes

Reviewer #6: Yes

3. Has the statistical analysis been performed appropriately and rigorously? 

Reviewer #1: Yes

Reviewer #2: Yes

Reviewer #4: (No Response)

Reviewer #5: Yes

Reviewer #6: Yes

4. Have the authors made all data underlying the findings in their manuscript fully available?

Reviewer #1: Yes

Reviewer #2: Yes

Reviewer #4: (No Response)

Reviewer #5: Yes

Reviewer #6: Yes

5. Is the manuscript presented in an intelligible fashion and written in standard English?

Reviewer #1: Yes

Reviewer #2: Yes

Reviewer #4: (No Response)

Reviewer #5: Yes

Reviewer #6: Yes

6. Review Comments to the Author

Reviewer #1: Regarding the manuscript entitled "The Synergistic Impact of Spirulina and Selenium Nanoparticles Mitigates the Adverse

Effects of Heat Stress on Semen Quality of Rabbits Bucks"

Abstract: Heat stress has a detrimental effect on animal fertility, particularly testicular functions,

including reduced sperm output and quality, which causes an economic loss in the

production of rabbits. The present trial investigated the efficacy of dietary Spirulina

(SP) (Arthrospira platensis), selenium nanoparticles (SeNPs), and their combination

(SP-SeNPs) on semen quality, haemato-biochemical, oxidative stress, immunity, and

sperm quality of heat-stressed (HS) rabbit bucks. Sixty mature bucks (APRI line) were

distributed into 6 groups of ten replicates under controlled conditions. Bucks in the 1st

group (control-NC) were kept under normal conditions (11-22 oC; 40 - 45% RH% =

relative humidity), while the 2nd group (control-HS) was kept under heat stress

conditions (32±0.50 °C; 60-66% RH %). The control groups were fed a commercial

pelleted diet and the other four heat-stressed groups were fed a commercial pelleted

diet with 1 g SP, 25 mg SeNPs, 1 g SP+25 mg SeNPs, and 1 g SP+50 mg SeNPs per

kg diet, respectively. The dietary inclusion of SP, SeNPs, and their combinations

significantly increased hemoglobin, platelets, total serum protein, high-density

lipoproteins, glutathione, glutathione peroxidase, superoxide dismutase, and seminal

plasma testosterone while decreased triglycerides, total cholesterol, urea, creatinine,

and malondialdehyde compared with the control-HS. Red blood cells, packed cell

volume, serum albumin, and testosterone significantly increased, while SeNPs,

SP+SeNPs25, and SP+SeNPs50 significantly decreased low-density lipoproteins,

aspartate, and alanine amino transferees. Total antioxidant capacity substantially

increased in serum and seminal plasma, while seminal plasma malondialdehyde

decreased in 25 or 50 mg of SeNPs+SP/kg groups. All supplements significantly

improved libido, sperm livability, concentration, intact acrosome, membrane integrity,

total output in fresh semen, and sperm quality in cryopreserved semen. SP-SeNPs50

had higher synergistic effect than SP-SeNPs25 on most different variables studied. In

conclusion, the dietary inclusion of SP plus SeNPs50 has a synergistic effect and is

considered a suitable dietary supplement for improving reproductive efficiency, health,

oxidative stress, and immunity of bucks in the breeding strategy under hot climates.

all comments have been addressed in the revised version nu authors.

Reviewer #2: (No Response)

Reviewer #4: Dear Authors,

I am pleased to inform you that I accepted your manuscript in the current form.

Regards

Reviewer #5: All of my suggestions were incorporated into the revised manuscript.

Reviewer #6: The following corrections need to be made:

Line 74….Se is a crucial component in …….Comment:… The Se is a crucial component in Response: Corrected as requested. Kindly check line 87.

My comment: Above correction was not done

Line 82-83: SeNPs and/or Spirulina (SP) were incorporated to mitigate the negative impacts on the heat-stressed growing rabbit by improving the antioxidant status. Comment: Rewrite this sentence to be the research hypothesis. Response: Done as requested. Kindly check lines 101-103.

My comment: L 98-99: The phrase “may affect the performance of male rabbits” is too general. Will the effect be positive or negative? What kind of performance in male rabbits will be affected?

7. PLOS authors have the option to publish the peer review history of their article (what does this mean?). If published, this will include your full peer review and any attached files.

Reviewer #1: No

Reviewer #2: No

Reviewer #4: No

Reviewer #5: **Yes: **Abdelazeem Sayed Abdelazeem

Reviewer #6: **Yes: **Kwaku Adomako

---

## [Author Response · Author response to Decision Letter 2]

28 Apr 2023

Responses to the comments of the Editor

I found your manuscript interesting and fit the journal's scope. It has also improved well after the second revision round, but there are still some revisions that need to be fixed before accepting this paper. 

Please make the modifications requested by the reviewers shown below (Reviewers, 2 and 6)

Response:

Thank you for your time and consideration. We have now changed this manuscript and inserted the required modifications with point-by-point responses to each of the reviewer’s comments. The corrected sentences were followed using Track Changes in the revised manuscript. 

Responses to the comments of Reviewer #2

All comments have been addressed. Please look the attached file

#Attached file:

delete antioxidant and. oxidative markers consist of both antioxidants and pro oxidants (MDA)

Response: Thank you for your comment. Corrected in Table 3 as follows:

“Oxidative markers in the blood”

separate markers and in

Response: Thank you for your comment. Corrected in Table 3 as follows:

“Oxidative markers in the blood”

SAME AS ABOVE.

Response: Thank you for your comment. Corrected in Table 3 as follows:

“Oxidative markers in the seminal plasma”

Responses to the comments of Reviewer #6

The following corrections need to be made: Line 74….Se is a crucial component in …….Comment:… The Se is a crucial component in Response: Corrected as requested. Kindly check line 87. My comment: Above correction was not done.

Response: Thank you for your careful reading that ensures the perfection of the manuscript. Corrected in line 83 as follows:

“The Se is a crucial component in the…….”

Line 82-83: SeNPs and/or Spirulina (SP) were incorporated to mitigate the negative impacts on the heat-stressed growing rabbit by improving the antioxidant status. Comment: Rewrite this sentence to be the research hypothesis. Response: Done as requested. Kindly check lines 101-103.

My comment: L 98-99: The phrase “may affect the performance of male rabbits” is too general. Will the effect be positive or negative? What kind of performance in male rabbits will be affected?

Response: We greatly appreciate your constructive comment. Corrected in line 99 as follows:

“we hypothesize that a combination of SP with different levels of SeNPs may positively affect the reproductive performance of male rabbits under HS.”

---

## [Decision Letter · Decision Letter 3]

22 May 2023

PONE-D-23-02023R3The Synergistic Impact of Spirulina and Selenium Nanoparticles Mitigates the Adverse Effects of Heat Stress on Semen Quality of Rabbits BucksPLOS ONE

Dear Dr. El Basuini,

Thank you for submitting your manuscript to PLOS ONE. After careful consideration, we feel that it has merit but does not fully meet PLOS ONE’s publication criteria as it currently stands. Therefore, we invite you to submit a revised version of the manuscript that addresses the points raised during the review process.

Dear Dr., Mohammed Fouad El Basuini

Thank you for submitting your manuscript to PLOS ONE. After careful consideration, we feel that it has merit but does not fully meet PLOS ONE’s publication criteria as it currently stands. Therefore, we invite you to submit a revised version of the manuscript that addresses the points raised during the review process.<o:p></o:p>

Thank you for submitting your manuscript to PLOS ONE. After careful consideration, we have decided that your manuscript needs Minor Revision.<o:p></o:p>

Kind regards,<o:p></o:p>

Prof. Lamiaa Mostafa Radwan, Ph.D.<o:p></o:p>

Academic Editor<o:p></o:p>

PLOS ONE<o:p></o:p>

Editor Comment<o:p></o:p>

Must be make the modifications requested by the reviewer 2.<o:p></o:p>

Please look the attached file and make all modified required. <o:p></o:p>

**Reviewer 1<o:p></o:p>**

All comments have been addressed<o:p></o:p>

**Reviewer 2<o:p></o:p>**

The authors did not address comments and review suggestions from my earlier review.

i cannot suggest the publications of this manuscript in its current form.**<o:p></o:p>**

Please submit your revised manuscript by Jul 06 2023 11:59PM.  If you will need more time than this to complete your revisions, please reply to this message or contact the journal office at plosone@plos.org. Please include the following items when submitting your revised manuscript:A rebuttal letter that responds to each point raised by the academic editor and reviewer(s). You should upload this letter as a separate file labeled 'Response to Reviewers'.A marked-up copy of your manuscript that highlights changes made to the original version. You should upload this as a separate file labeled 'Revised Manuscript with Track Changes'.An unmarked version of your revised paper without tracked changes. You should upload this as a separate file labeled 'Manuscript'.If applicable, we recommend that you deposit your laboratory protocols in protocols.io to enhance the reproducibility of your results. Protocols.io assigns your protocol its own identifier (DOI) so that it can be cited independently in the future. For instructions see: https://journals.plos.org/plosone/s/submission-guidelines#loc-laboratory-protocols. Additionally, PLOS ONE offers an option for publishing peer-reviewed Lab Protocol articles, which describe protocols hosted on protocols.io. Read more information on sharing protocols at https://plos.org/protocols?utm_medium=editorial-email&utm_source=authorletters&utm_campaign=protocols.

We look forward to receiving your revised manuscript.

Kind regards,

Lamiaa Mostafa Radwan, Ph.D.

Academic Editor

PLOS ONE

Journal Requirements:

Additional Editor Comments:

Dear Dr., Mohammed Fouad El Basuini

Thank you for submitting your manuscript to PLOS ONE. After careful consideration, we feel that it has merit but does not fully meet PLOS ONE’s publication criteria as it currently stands. Therefore, we invite you to submit a revised version of the manuscript that addresses the points raised during the review process.

Thank you for submitting your manuscript to PLOS ONE. After careful consideration, we have decided that your manuscript needs Minor Revision.

Kind regards,

Prof. Lamiaa Mostafa Radwan, Ph.D.

Academic Editor

PLOS ONE

Editor Comment

Must be make the modifications requested by the reviewer 2.

Please look the attached file and make all modified required.

Reviewer 1

All comments have been addressed

Reviewer 2

The authors did not address comments and review suggestions from my earlier review.

i cannot suggest the publications of this manuscript in its current form.

Reviewers' comments:

Reviewer's Responses to Questions

**Comments to the Author**

1. If the authors have adequately addressed your comments raised in a previous round of review and you feel that this manuscript is now acceptable for publication, you may indicate that here to bypass the “Comments to the Author” section, enter your conflict of interest statement in the “Confidential to Editor” section, and submit your "Accept" recommendation.

Reviewer #2: (No Response)

Reviewer #6: All comments have been addressed

2. Is the manuscript technically sound, and do the data support the conclusions?

Reviewer #2: Partly

Reviewer #6: Yes

3. Has the statistical analysis been performed appropriately and rigorously? 

Reviewer #2: N/A

Reviewer #6: Yes

4. Have the authors made all data underlying the findings in their manuscript fully available?

Reviewer #2: Yes

Reviewer #6: Yes

5. Is the manuscript presented in an intelligible fashion and written in standard English?

Reviewer #2: No

Reviewer #6: Yes

6. Review Comments to the Author

Reviewer #2: The authors did not address comments and review suggestions from my earlier review.

i cannot suggest the publications of this manuscript in its current form.

Reviewer #6: (No Response)

7. PLOS authors have the option to publish the peer review history of their article (what does this mean?). If published, this will include your full peer review and any attached files.

Reviewer #2: **Yes: **Dr Jimoh Olatunji Abubakar

Reviewer #6: **Yes: **Kwaku Adomako

---

## [Author Response · Author response to Decision Letter 3]

23 May 2023

Responses to the comments of Editor

Comment: Must be make the modifications requested by the reviewer 2. Please look the attached file and make all modified required.

Response: All comments have been addressed as suggested. Kindly check the revised version with track changes.

Responses to the comments of Reviewer #2

Comment: The authors did not address comments and review suggestions from my earlier review. I cannot suggest the publications of this manuscript in its current form.

Response: Thank you for your time and insightful comments. All comments have been addressed as suggested. Please check the revised version with track changes.

#Attached file:

Re-arrange your references. 2-5 cannot come before 1-2

Response: Done as suggested and citations start with No. 1. Please check line 54.

biomarkers

Response: Done as suggested. Kindly check Table 3.

delete antioxidant and. oxidative markers consist of both antioxidants and pro oxidants (MDA)

Response: Thank you for your comment. Corrected in Table 3 as follows:

“Oxidative markers in the blood”

separate markers and in

Response: Thank you for your comment. Corrected in Table 3 as follows:

“Oxidative markers in the blood”

SAME AS ABOVE.

Response: Thank you for your comment. Corrected in Table 3 as follows:

“Oxidative markers in the seminal plasma”

This caption in the title is not a true reflection of what was done. heamato-biochemical, semen quality, testosterone, oxidative status and sperm characteristics were all reported in this study. is suggest the use of physiology in place of semen quality in the title or list the above indices in the title, if not too lengthy

Response: Done as suggested as follows (Please check the title):

The Synergistic Impact of Spirulina and Selenium Nanoparticles Mitigates the Adverse Effects of Heat Stress on the Physiology of Rabbits Bucks

reference 2-5 cant start the manuscript. Re-number from 1.

Response: Done as suggested and citations start with No. 1. Please check line 54.

initial corrections pointed out in the earlier review are not addressed.

Response: Done as suggested. Kindly check Table 3.

oxidative markers are a combination of antioxidant and prooxidant indicators. hence oxidative markers will suffice.

Response: Done as suggested. Kindly check Table 3.

---

## [Decision Letter · Decision Letter 4]

12 Jun 2023

The Synergistic Impact of Spirulina and Selenium Nanoparticles Mitigates the Adverse Effects of Heat Stress on the Physiology of Rabbits Bucks

PONE-D-23-02023R4

Dear Dr. Mohammed Fouad El Basuini

We’re pleased to inform you that your manuscript has been judged scientifically suitable for publication and will be formally accepted for publication once it meets all outstanding technical requirements.

Kind regards,

Lamiaa Mostafa Radwan, Ph.D.

Academic Editor

PLOS ONE

Additional Editor Comments (optional):

That manuscript was reviewed by 4 reviewers. Three reviewers gave accept for publication, and now the reviewer 4 has given Accept

Reviewers' comments:

Reviewer's Responses to Questions

**Comments to the Author**

1. If the authors have adequately addressed your comments raised in a previous round of review and you feel that this manuscript is now acceptable for publication, you may indicate that here to bypass the “Comments to the Author” section, enter your conflict of interest statement in the “Confidential to Editor” section, and submit your "Accept" recommendation.

Reviewer #2: All comments have been addressed

2. Is the manuscript technically sound, and do the data support the conclusions?

Reviewer #2: Yes

3. Has the statistical analysis been performed appropriately and rigorously? 

Reviewer #2: Yes

4. Have the authors made all data underlying the findings in their manuscript fully available?

Reviewer #2: Yes

5. Is the manuscript presented in an intelligible fashion and written in standard English?

Reviewer #2: Yes

6. Review Comments to the Author

Reviewer #2: The manuscript has been squarely improved. however, little amendments are required as poited out in the reviewed file

7. PLOS authors have the option to publish the peer review history of their article (what does this mean?). If published, this will include your full peer review and any attached files.

Reviewer #2: **Yes: **Dr Jimoh Olatunji Abubakar

---

## [Editor Report · Acceptance letter]

4 Jul 2023

PONE-D-23-02023R4 

The Synergistic Impact of Spirulina and Selenium Nanoparticles Mitigates the Adverse Effects of Heat Stress on the Physiology of Rabbits Bucks 

Dear Dr. El Basuini:

I'm pleased to inform you that your manuscript has been deemed suitable for publication in PLOS ONE. Congratulations! Your manuscript is now with our production department. 

Kind regards, 

on behalf of

Prof. Dr. Lamiaa Mostafa Radwan 

Academic Editor

PLOS ONE